# Protein acetylation affects acetate metabolism, motility and acid stress response in *Escherichia coli*

Sara Castaño-Cerezo[1], Vicente Bernal[1,*], Harm Post[2,3], Tobias Fuhrer[4], Salvatore Cappadona[2], Nerea C Sánchez-Díaz[1], Uwe Sauer[4], Albert JR Heck[2,3], AF Maarten Altelaar[2,3] & Manuel Cánovas[1,**]

## Abstract

Although protein acetylation is widely observed, it has been associated with few specific regulatory functions making it poorly understood. To interrogate its functionality, we analyzed the acetylome in *Escherichia coli* knockout mutants of *cobB*, the only known sirtuin-like deacetylase, and *patZ*, the best-known protein acetyltransferase. For four growth conditions, more than 2,000 unique acetylated peptides, belonging to 809 proteins, were identified and differentially quantified. Nearly 65% of these proteins are related to metabolism. The global activity of CobB contributes to the deacetylation of a large number of substrates and has a major impact on physiology. Apart from the regulation of acetyl-CoA synthetase, we found that CobB-controlled acetylation of isocitrate lyase contributes to the fine-tuning of the glyoxylate shunt. Acetylation of the transcription factor RcsB prevents DNA binding, activating flagella biosynthesis and motility, and increases acid stress susceptibility. Surprisingly, deletion of *patZ* increased acetylation in acetate cultures, which suggests that it regulates the levels of acetylating agents. The results presented offer new insights into functional roles of protein acetylation in metabolic fitness and global cell regulation.

**Keywords** flagella biosynthesis; isocitrate lyase; metabolic regulation; sirtuin
**Subject Categories** Post-translational Modifications, Proteolysis & Proteomics; Microbiology, Virology & Host Pathogen Interaction
**Mol Syst Biol.** (2014) 10: 762

## Introduction

From bacteria to higher animals and plants, organisms need to adapt to their environment. Physiological processes are regulated at several levels, from transcriptional control of gene expression to allosteric effects on enzyme activities. Reversible post-translational modification is a fast mechanism for controlling the activity of proteins, and in particular, they are expected to have key relevance in metabolism, controlling the use of competing pathways (Heinemann & Sauer, 2010; Gerosa *et al*, 2013). Almost 200 different types of protein modifications have been described, with lysine residues harboring most types of modifications on its side chain (Hershko & Ciechanover, 1998; Martin & Zhang, 2005; Kai *et al*, 2010; Weinert *et al*, 2013a).

Although protein acetylation at lysine residues has been known to occur since the 1960s, it has emerged in the past 10 years as a highly prominent post-translational modification, widely spread in all domains of life. Conventionally, this reversible protein modification has been mainly linked to transcriptional regulation: Increased acetylation of histones decreases its interaction with DNA, thereby decreasing nucleosome compactness (Sterner & Berger, 2000). Recent studies have shown that a high percentage of proteins related to metabolism are lysine-acetylated (Yu *et al*, 2008; Zhang *et al*, 2009, 2012; Wang *et al*, 2010; Henriksen *et al*, 2012; Lundby *et al*, 2012; Van Noort *et al*, 2012; Kim *et al*, 2013). The bacterial paradigm of this regulation is *Salmonella enterica*, where acetyl-coenzyme A synthetase was the first enzyme whose activity was described to be reversibly regulated by lysine acetylation (Starai *et al*, 2002; Starai & Escalante-Semerena, 2004). In this microorganism, almost 200 proteins have been reported to be acetylated, and almost half of these targets are metabolic enzymes. Notwithstanding its widespread appearance, a thorough characterization of the functional implications that protein acetylation has on the bacterial physiology and, in particular, its metabolism is still largely lacking.

The enzymes involved in protein acetylation and deacetylation have already been described in several bacteria (Starai *et al*, 2002; Zhao *et al*, 2004; Gardner *et al*, 2006; Gardner & Escalante-Semerena, 2009; Crosby *et al*, 2010, 2012b; Nambi *et al*, 2010; Tucker & Escalante-Semerena, 2010; Mikulik *et al*, 2012; Hayden *et al*, 2013; Ho Jun *et al*, 2013). CobB, the first bacterial sirtuin-like deacetylase described, is capable of deacetylating acetyl-lysine residues using $NAD^+$ as a substrate (Tsang & Escalante-Semerena, 1998). In *Bacillus subtilis*, $NAD^+$-dependent and independent deacetylases coexist (Gardner *et al*, 2006; Gardner & Escalante-Semerena, 2009).

1  Departamento de Bioquímica y Biología Molecular B e Inmunología, Facultad de Química, Universidad de Murcia, Campus de Excelencia Mare Nostrum, Murcia, Spain
2  Biomolecular Mass Spectrometry and Proteomics Group, Bijvoet Center for Biomolecular Research and Utrecht Institute for Pharmaceutical Sciences, Utrecht University, Utrecht, The Netherlands
3  Netherlands Proteomics Center, Utrecht, The Netherlands
4  Institute of Molecular Systems Biology, ETH Zurich, Zurich, Switzerland
   *Corresponding author. Tel: +34 620 308626; E-mail: vicente.bernal@gmail.com
   **Corresponding author. Tel: +34 868 887393; E-mail: mcanovas@um.es

A Gnc5-like acetyltransferase, Pat, that uses acetyl-CoA as substrate was discovered in *S. enterica* (Starai & Escalante-Semerena, 2004). The only known lysine acetyltransferase in *Mycobacterium* is allosterically activated by cyclic AMP, and this metabolite is the main activator of virulence in these species (Nambi *et al*, 2010). It remains unclear whether other acetyltransferases and deacetylases exist in these microorganisms (Gardner *et al*, 2006; Nambi *et al*, 2010; Tucker & Escalante-Semerena, 2010; Crosby *et al*, 2012a).

Several recent proteomic studies have revealed that lysine acetylation is abundant in *E. coli*, the most recent one reported over 1,000 acetylated proteins (Yu *et al*, 2008; Zhang *et al*, 2009, 2012; Weinert *et al*, 2013b). Physiological implications of lysine acetylation in *E. coli* are subject of intense study and include altered activity of an array of different cellular machineries such as the acetyl-CoA synthetase, RNA polymerase, the chemotaxis response regulator (CheY), the regulator of capsule synthesis (RcsB), ribonuclease R (RNase R) and N-hydroxyarylamine O-acetyltransferase (Li *et al*, 2010; Thao *et al*, 2010; Castaño-Cerezo *et al*, 2011; Lima *et al*, 2012; Hu *et al*, 2013; Zhang *et al*, 2013; Bernal *et al*, 2014). Despite this recent progress, still relatively little is known about the physiological relevance of widespread protein lysine acetylation and how it is, in turn, affected by environmental conditions.

Here, we set out to understand how the protein acetylation state impacts the physiology of *E. coli*. We use for that knockout strains of *cobB* and *patZ*. The sirtuin CobB is the only deacetylase known in *E. coli*. Interestingly, the expression of the best-known acetyltransferase PatZ (formerly, YfiQ) is affected by metabolic signals (Castaño-Cerezo *et al*, 2011). Pathways affected by protein acetylation were identified in Δ*cobB* and Δ*patZ* mutant strains taking a systems approach utilizing dedicated proteomic and transcriptomic tools. Complementary, metabolic flux analysis and molecular biology studies were focused on the regulation of the central carbon metabolism (especially the acetate overflow and glyoxylate shunt routes) and signaling pathways related to chemotaxis and stress response.

# Results

In order to demonstrate how protein acetylation impacts the physiology of *E. coli*, we analyzed the effects of the deletion of its sole (known) lysine deacetylase (*cobB*) and its best characterized lysine acetyltransferase (*patZ*). Our experiments were performed under physiologically relevant conditions, selecting those where large changes in protein acetylation were expected. We have previously demonstrated that the expression of *patZ* is up-regulated by cAMP (e.g. upon glucose limitation or during growth on non-PTS carbon sources) (Castaño-Cerezo *et al*, 2011). The growth of the three *E. coli* strains was compared in glucose batch (non-carbon-limited) and chemostat (carbon-limited) cultures and in acetate (non-PTS carbon source) batch cultures. It was under gluconeogenic conditions, such as the acetate batch and carbon-limited chemostat cultures, that the phenotype of the Δ*cobB* mutant was most affected (Supplementary Fig S1). The severe growth impairment of the Δ*cobB* mutant during conditions of high expression of the *patZ* acetyltransferase led us to hypothesize that this effect was caused by increased lysine acetylation of proteins crucial for optimal growth.

## Mapping lysine-acetylated proteins in *Escherichia coli*

The profound phenotypic effects observed in the mutants could indicate that protein lysine acetylation patterns were altered (Table 1) (Castaño-Cerezo *et al*, 2011). In order to prove this, a proteomic study was carried out to identify and quantify acetylated peptides and relate changes in acetylation to observed physiological changes. Immunoprecipitation of acetylated peptides followed by high-resolution MS-based proteomics was performed: Stable isotope dimethyl labeling was used for the relative quantification of peptide acetylation ratios between strains (Boersema *et al*, 2009; Choudhary *et al*, 2009).

For each condition, four biological replicates were analyzed. Overall, 2,502 acetylated peptides were detected belonging to 809 different proteins (Supplementary Table S1). Approximately half of these proteins were acetylated on a single residue, while over 20% of the observed proteins were highly acetylated (i.e. modified in more than three sites) (Supplementary Fig S2). We quantified the relative ratio of peptide acetylation in Δ*cobB* and Δ*patZ* mutants compared to the wild-type under four different environmental conditions. As expected, the phenotypes of the mutants mirrored altered peptide acetylation ratios of proteins as detailed below (Supplementary Dataset S1).

The mutant Δ*patZ*, deficient in the best-known lysine acetyltransferase, did not lead to many changes in acetylation ratios in glucose cultures (Supplementary Fig S3A and C). Although a lower acetylation status was expected in this mutant, this was only the case in the chemostat cultures, where the abundance of almost 7% of the identified acetylated peptides was at least half compared to the wild-type (Fig 1A). However, none of the proteins with decreased acetylation levels have been demonstrated previously to be PatZ substrates. The acetylation ratios of the Δ*patZ* mutant in exponential phase glucose cultures were hardly altered compared with the wild-type (Supplementary Fig S3A), nor was its phenotype under this growth condition. It could be argued that, since in exponential phase the *patZ* gene expression is low, there should not be many differences in acetylation ratios in the Δ*patZ* mutant. However, also in glucose-limited chemostat cultures, the acetylation ratios did not show the expected trend (Fig 1A). This might be explained by the presence of at least 25 putative acetyltransferases existing in *E. coli* (Lima *et al*, 2012) that could potentially take over the function of PatZ. Surprising changes in the ratio of peptide acetylation were observed in the acetate cultures and, to lesser extent, in the stationary-phase glucose cultures in the Δ*patZ* mutant (Fig 1C; Supplementary Fig S3C). Deletion of *patZ* gene in acetate cultures led to an overall increased acetylation of the whole proteome: The acetylation ratio of 75% of peptides was more than twice that of the wild-type in acetate cultures (Fig 1C). Despite the high impact of *patZ* deletion on protein acetylation, no evident phenotypic effects were observed.

Deletion of *cobB* caused substantial phenotypic changes. The growth rate of this mutant was reduced in all conditions assayed (Table 1). Deletion of the only deacetylase known in *E. coli* should increase the degree of acetylation of proteins. Our results confirmed that CobB has a major role as deacetylase in *E. coli*. Over 17% of the acetylated peptides quantified showed increased acetylation under each of the chosen growth conditions (at least twofold in the Δ*cobB* mutant compared to the wild-type) (Fig 1B and D;

Table 1. Physiological characterization of *Escherichia coli* and its knockout mutants grown in glucose batch and glucose-limited chemostat cultures.

| | Glucose batch cultures | | | | |
|---|---|---|---|---|---|
| Physiology | wt | Δ*cobB* | Δ*patZ* | Δ*cobB*Δ*aceK* | Δ*patZ*Δ*aceK* |
| $\mu_{max}$ (h$^{-1}$) | $0.74 \pm 0.02$ | $0.68 \pm 0.03$ | $0.70 \pm 0.01$ | $0.65 \pm 0.01$ | $0.67 \pm 0.01$ |
| $q_{glc}$ [mmol/g/h] | $-8.87 \pm 1.19$ | $-10.51 \pm 1.74$ | $-8.20 \pm 0.16$ | $-8.21 \pm 0.44$ | $-8.36 \pm 1.79$ |
| $q_{acet}$ [mmol/g/h] | $4.42 \pm 0.17$ | $5.98 \pm 0.22$ | $3.97 \pm 1.03$ | $6.07 \pm 0.15$ | $4.49 \pm 0.11$ |
| $Y_{cel/glc}$ (g/g) | $0.46 \pm 0.03$ | $0.36 \pm 0.02$ | $0.47 \pm 0.01$ | $0.44 \pm 0.02$ | $0.45 \pm 0.04$ |
| | Glucose chemostat cultures | | | | |
| Physiology | wt | Δ*cobB* | Δ*patZ* | Δ*cobB*Δ*aceK* | Δ*patZ*Δ*aceK* |
| $\mu_{max}$ (h$^{-1}$) | $0.23 \pm 0.01$ | $0.19 \pm 0.01$ | $0.23 \pm 0.01$ | $0.19 \pm 0.01$ | $0.23 \pm 0.01$ |
| $q_{glc}$ [mmol/g/h] | $-2.70 \pm 0.17$ | $-3.68 \pm 0.23$ | $-2.64 \pm 0.09$ | $-2.55 \pm 0.01$ | $-2.85 \pm 0.28$ |
| $q_{acet}$ [mmol/g/h] | $0.00 \pm 0.00$ | $0.63 \pm 0.14$ | $0.00 \pm 0.00$ | $0.62 \pm 0.03$ | $0.00 \pm 0.00$ |
| $Y_{cel/glc}$ (g/g) | $0.47 \pm 0.02$ | $0.29 \pm 0.02$ | $0.48 \pm 0.03$ | $0.41 \pm 0.01$ | $0.45 \pm 0.03$ |

Supplementary Fig S3B and D). The number of peptides with increased acetylation was higher in the conditions where the change in phenotype was more profound, that is, acetate and chemostat cultures (30 and 21%, respectively). Since the expression of an inactivated CobB protein, with a mutation in its catalytic H110 residue, did not rescue the phenotype of the *cobB* knockout mutant, we concluded that the phenotypic and proteomic effect observed in this mutant is caused by the absence of the deacetylase activity (Supplementary Fig S4).

The intriguing accumulation of acetylated proteins in acetate cultures was further analyzed. About 15% of peptide acetylation ratios were significantly different in the two mutants (Fig 2A). Statistical significance levels were determined by two-sample *t*-test, followed by multiple testing correction using permutation-based FDR < 0.05 (Tusher *et al*, 2001) (Supplementary Dataset S2). These differences in the protein acetylation profiles of the *patZ* and *cobB* mutants in acetate mirror the different degree of phenotype alteration observed. To get an insight on which of the acetylated proteins might be responsible for the different phenotypes, we focused on those with high acetylation ratios only in the *cobB* mutant. All acetylated peptides with a ratio twofold higher than the wild-type were analyzed; of these, 25 peptides were identified as highly acetylated only in the *cobB* mutant but not in the *patZ* mutant (two-sample *t*-test, adjusted for multiple testing using permutation-based FDR < 0.05) (Fig 2B; Supplementary Dataset S2). Proteins in this group which were acetylated at lysine residues which have been previously identified as relevant for their function were identified by manual curation (Fig 2C; Supplementary Table S2). These proteins could contribute to growth impairment in the *cobB* mutant. On the other hand, the higher acetylation levels observed in the Δ*patZ* mutant were probably caused by a deregulation of chemical acetylation. This may not alter cell growth significantly due to: (i) its lower specificity (probably affecting only a fraction of the total cellular protein), and (ii) due to the presence of an active CobB, which would deacetylate those proteins which are its true substrates and whose acetylation state is really crucial for cellular functions.

Acetylated sequences identified were analyzed in order to find an acetylation consensus motif (Fig 3A). This motif was similar to those previously reported by other authors (Weinert *et al*, 2013b).

The glycine residue at position −1 is known to be conserved (Crosby *et al*, 2012a; Crosby & Escalante-Semerena, 2014), and, as previously reported, acetylated lysines are more likely found near other lysines, thus decreasing the length of the tryptic peptides (Choudhary *et al*, 2009; Weinert *et al*, 2011, 2013b). Another characteristic of the acetylation motif in *E. coli* is the high abundance of aspartic and glutamic residues close to the acetylated lysine, which has also been observed in the acetylome of rat and *Thermus thermophilus* (Lundby *et al*, 2012; Okanishi *et al*, 2013).

Importantly, detection of acetylated peptides is affected by protein abundance in the cell since minority proteins are likely out of the reach of current techniques. Therefore, the subset of acetylated proteins detected is biased toward those proteins whose concentration in the cell is high (Fig 3B; Supplementary Materials and Methods). The analysis of the functions of acetylated proteins (Gene Ontology terms) sheds light on the major biological processes affected. In our study, 64% of the modified proteins detected have a metabolic function, and almost 80% of these are involved in primary metabolism, for example, nucleotide and amino acid biosynthesis and carbohydrate metabolism (Supplementary Fig S5). Other represented functions in our dataset relate to sensing and stimulus responses, mostly proteins belonging to two component systems, such as ArcA, RcsB, CpxR and EvgA. Additionally, almost 7% of the acetylated proteins have a role in transcription (Supplementary Table S3). However, the frequency of these GO terms in the set of acetylated proteins reflects their own frequency in the whole genome, which means that no specific function is overrepresented in the group of acetylated proteins and reveals that protein acetylation occurs on every type of protein independently of its function (Supplementary Materials and Methods).

Changes in the acetylation profiles are insufficient to infer regulatory roles. In an attempt to identify physiological roles of lysine acetylation, we focused on the pathways that were quantitatively most affected, driven by the physiology, gene expression and protein acetylation profiles of the mutants. The phenotype of the Δ*cobB* mutant was most affected in the acetate batch and carbon-limited chemostat cultures. In fact, inefficient growth and a clear shift in the acetate overflow were evident in carbon-limited chemostat cultures (Supplementary Fig S1), which

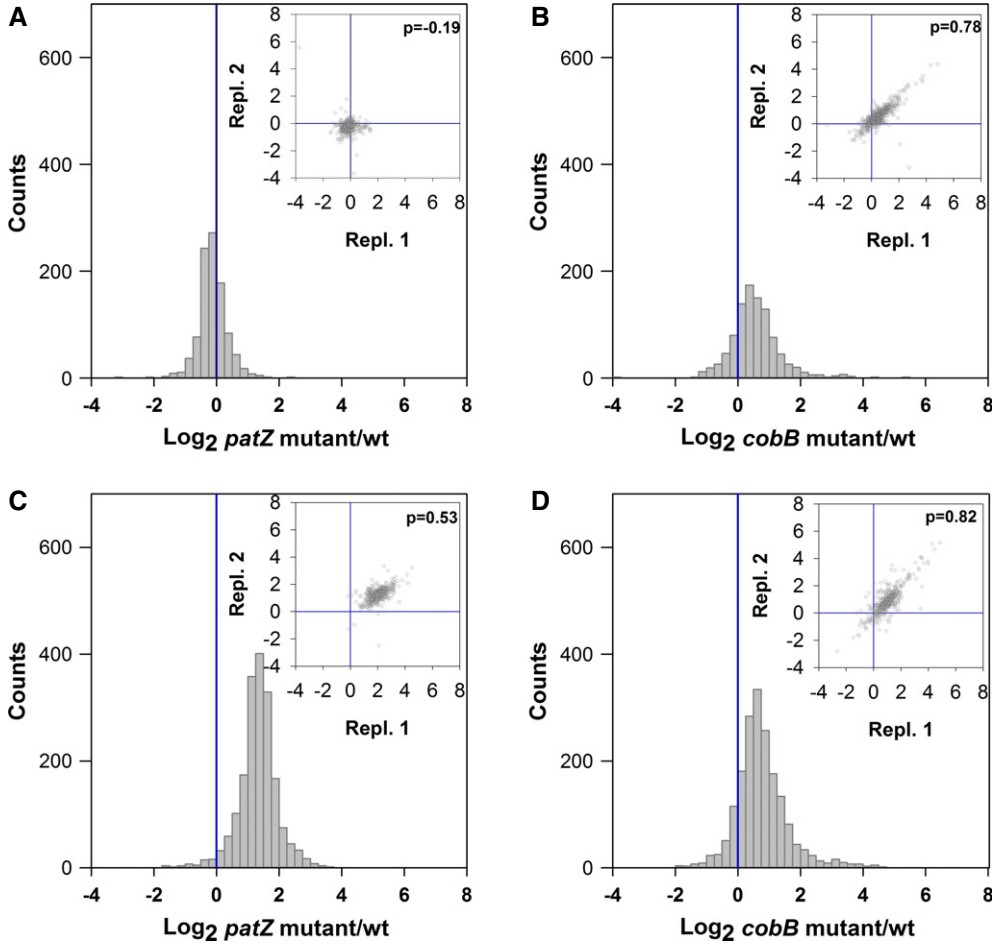

**Figure 1.** Frequency histogram of the acetylated peptide ratios (log$_2$) of *Escherichia coli* Δ*patZ* and Δ*cobB* mutants referred to the wild-type strain.

A–D    Bacteria were grown in minimal media: glucose-limited chemostat cultures (A, B) and batch acetate cultures (C, D). Acetylation data are expressed as the ratios for Δ*patZ*/wt (A, C) and Δ*cobB*/wt (B, D) strains. Frequency histograms represent the median of the log$_2$ ratios from four biological replicates. Insert figures represent the correlation of acetylated peptide ratios in two biological replicates (Pearson's correlation of each condition/mutant comparison is shown on the plot).

Source data are available online for this figure.

led us to investigate the role of acetylation on the regulation of the two pathways that are essential for acetate assimilation: acetyl-CoA synthetase and the glyoxylate shunt. In addition, gene expression profiles underlined a clear effect on the motility and acid stress genes, which are both belonging to the RcsB regulon. In the following sections, major implications of protein lysine acetylation in these pathways will be dissected.

### The relationship between protein acetylation and acetate metabolism in *Escherichia coli*

In glucose-limited chemostat cultures, acetate overflow is a function of the dilution rate due to catabolite repression of the acetyl-CoA synthetase encoding gene (*acs*) (Vemuri *et al*, 2006; Valgepea *et al*, 2010; Renilla *et al*, 2012). The Δ*cobB* strain produces acetate in low dilution rate glucose chemostat cultures; in fact, it has a phenotype similar to the Δ*acs* mutant (Supplementary Fig S1), where yield is limited by its inability to scavenge overflown acetate (Renilla *et al*, 2012). The importance of the de-acetylation of this enzyme for

acetate metabolization has been demonstrated in a previous study, where the Δ*cobB* mutant could not grow efficiently in acetate (Castaño-Cerezo *et al*, 2011). However, the reduced growth rate and biomass yield in acetate batch and glucose-limited chemostat cultures cannot be explained by the modification of one single enzyme, and, most likely, the phenotypes observed are the result of more profound effects exerted on other pathways essential for acetate assimilation.

To further demonstrate the effect of acetylation on acetate metabolism, the deacetylation of acetyl-CoA synthetase by CobB was demonstrated *in vitro*. This enzyme is more active in its deacetylated form: Activity increased 40 times after incubation with CobB compared to the control without the deacetylase or with the CobB inhibitor nicotinamide (NAM). Deacetylation was confirmed by Western blotting (Fig 4A). In these *in vitro* assays, CobB-catalyzed deacetylation of K609 and K617 was detected by LC-MS/MS (Fig 4B). The involvement of conserved K609 reversible acetylation on activity has been previously demonstrated in *Salmonella enterica* and other bacteria (Starai *et al*, 2002, 2005).

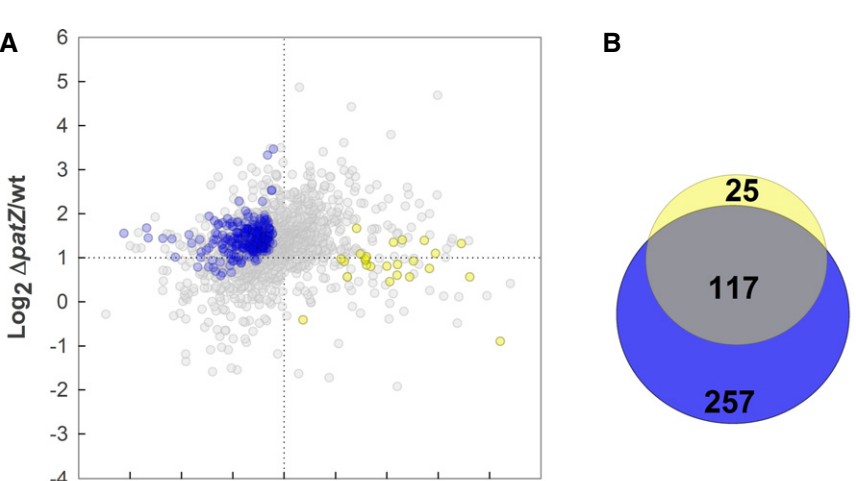

**C**

| Gene name | Uniprot | Acetyl(K) | Log₂ ΔcobB/wt | p-value | Reference |
|-----------|---------|-----------|----------------|---------|-----------|
| *fbp* | P0A993 | 269 | 1.37 | 0.013 | (El-Magharbi *et al*, 1992) |
| *rpoD* | P00579 | 557 | 4.6 | 0.003 | (Paget & Helmann, 2003) |
| *rai* | P0AD49 | 87 | 3.4 | 0.002 | (Agafonov *et al*, 1999), (Maki *et al*, 2000) |
| *gapA* | P0A9B2 | 184 | 2.14 | n.d. | (Kuhn *et al*, 2014) |
| *fadD* | P694513 | 543 | 1.87 | n.d. | (Crosby *et al*, 2010) |

**Figure 2.    Differential protein acetylation in *patZ* and *cobB* mutants in acetate cultures.**

A    Representation of the log₂ acetylation ratio for ΔpatZ/wt (*y*-axis) and ΔcobB/wt (*x*-axis) of peptides in acetate cultures. Values used were obtained from four biological replicates (*n* = 1,875 different acetylated peptides detected). Peptides with significantly different ratios in the *patZ* and *cobB* mutants are colored in blue and yellow, respectively (two-sample *t*-test, adjusted for multiple testing using permutation-based FDR < 0.05).

B    Venn diagram showing the overlap of significant acetylated peptides with an acetylation ratio compared with the wild-type strain higher than 2 (FDR < 0.05). In blue are represented the number of significant acetylated peptides of ΔpatZ and in yellow those for ΔcobB mutant. The overlapping grey region represents those peptides which are significantly acetylated in both mutants.

C    Representative examples of proteins acetylated in the ΔcobB mutant in acetate cultures, but not altered in the *patZ* mutant, whose function is probably affected. Further information is detailed in the main text and in the Supplementary Materials and Methods.

Source data are available online for this figure.

The increase of acetylation of acetyl-CoA synthetase was reflected by *in vivo* activity levels. Acetyl-CoA synthetase activity in the ΔcobB mutant is almost half of that of the wild-type in glucose-limited chemostat cultures and four times lower in acetate cultures (Supplementary Fig S6). The peptide-containing K609 was found in the wild-type strain and the ΔcobB mutant but was not present in the ΔpatZ mutant, which supports that PatZ specifically acetylates acetyl-CoA synthetase (Supplementary Fig S7). Eight additional acetylation sites were found in acetyl-CoA synthetase in chemostat and acetate cultures, but none of their acetylation ratios showed significant changes in the *cobB* mutant, suggesting that they cannot be deacetylated by CobB (Supplementary Dataset S3).

The reduced growth rate and biomass yield of the ΔcobB mutant under acetate batch and glucose-limited chemostat conditions indicate that another acetate assimilation pathway could be affected. We hypothesized that the glyoxylate shunt might be affected by increased acetylation in the ΔcobB mutant, as this pathway is essential for growth on acetate as the sole carbon source and

contributes to glucose catabolism in glucose-limited culture (Fischer & Sauer, 2003). To further explore the functional consequences of the acetylation of metabolic enzymes, intracellular fluxes were determined by ¹³C experiments (Supplementary Table S4). Flux through the glyoxylate shunt decreased by 34% in the ΔcobB mutant in glucose chemostat cultures (Fig 4C). The isocitrate node is an important regulation point for anabolism and catabolism. Isocitrate lyase (glyoxylate shunt) and isocitrate dehydrogenase (TCA cycle) compete for their common substrate, and fluxes through the node are regulated by reversible phosphorylation of isocitrate dehydrogenase (LaPorte & Koshland, 1983; Borthwick *et al*, 1984; LaPorte *et al*, 1984). It has been proposed in *S. enterica* that this metabolic node is controlled by the acetylation of the bifunctional isocitrate dehydrogenase phosphatase/kinase AceK (Wang *et al*, 2010). However, acetylation of AceK was not detected in our proteomic study. Also, quantification of metabolic fluxes in the ΔcobBΔaceK and ΔpatZΔaceK mutants demonstrated that lysine acetylation was not affecting AceK function (see Supplementary Materials and Methods for further information).

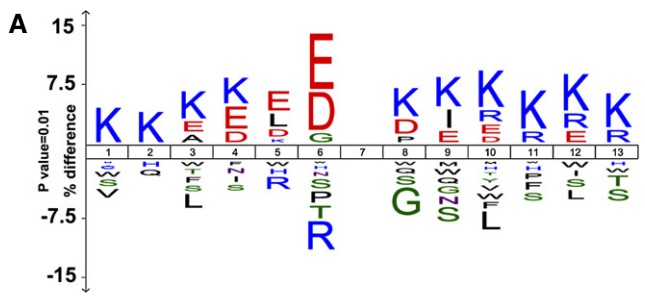

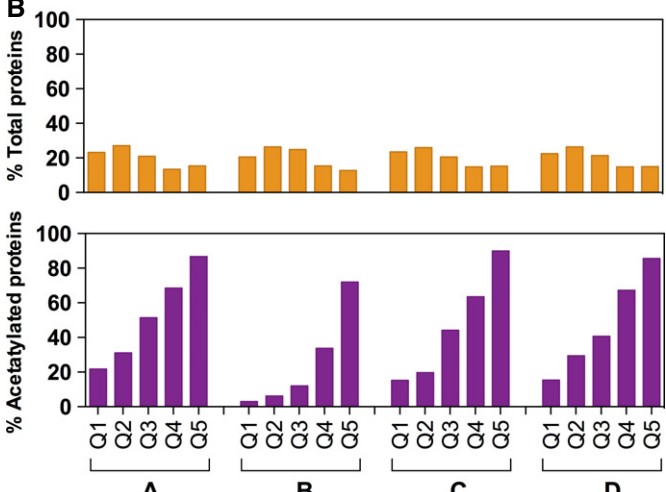

**Figure 3.  Analysis of acetylated proteins: acetylation motif and effect of protein abundance on acetylation.**

A   Sequence motif surrounding acetylated lysines. Logo was created using the Icelogo software package. All acetylated peptides found with an acetyl (K) probability higher than 0.9 and a cutoff *P*-value < 0.01 were used.

B   Frequency of protein acetylation detection as a function of protein abundance in the cell. Proteins in whole-cell extract protein digests were analyzed by LC-MS. Quantified proteins were sorted as a function of their relative abundance into five quantiles Q1–Q5. The less abundant proteins belong to Q1, and the more abundant ones to Q5. This analysis was performed for each experimental replicate and in all conditions assayed in this study: A, acetate cultures; B, chemostat cultures; C, glucose batch cultures exponential phase; D, glucose batch cultures stationary phase (*n* = 4 per condition). Upper bar plot: Orange bars represent the percentage of total proteins detected belonging to each quantile at each condition (*n* = 4). Lower bar plot: Purple bars represent the percentage of acetylated proteins belonging to each protein quantile. Further information is detailed in the Supplementary Materials and Methods.

Source data are available online for this figure.

Once demonstrated that deficient functioning of the glyoxylate shunt was not due to the acetylation of AceK, the acetylation of other enzymes involved in the pathway was investigated. Clear evidence for the acetylation of isocitrate lyase was presented in the proteomic data, with thirteen acetylation sites identified of which five sites showed increased acetylation levels (K13, K34, K308, K326 and K331) in the ΔcobB mutant compared to the wild-type strain in chemostat cultures, and one site (K308) in acetate cultures (Supplementary Dataset S3). *In vitro* isocitrate lyase deacetylation assays showed that after the incubation with the sirtuin-like deacetylase CobB, the activity of AceA increased around 40% (Fig 4D). We have

found that CobB deacetylates lysines 13 and 308 extensively, a reaction which is inhibited by nicotinamide (Fig 4E). This result is in agreement with previous finding that the acetylation of lysine 308 decreases AceA activity in *S. enterica* (Wang *et al*, 2010).

*In vivo*, relative quantification of the protein expression patterns in the different conditions revealed changes in protein levels in the ΔcobB mutant. Glyoxylate shunt proteins were less abundant in the ΔcobB mutant (approximately 50% of the levels observed in the wild-type) (Supplementary Fig S8; Supplementary Dataset S4). Finally, the enzyme activities of isocitrate lyase and isocitrate dehydrogenase were measured (Supplementary Fig S6). In acetate cultures and glucose-limited cultures, the activity of isocitrate lyase was lower in the ΔcobB mutant compared to the wild-type strain. This was especially true in chemostat cultures, where it decreased almost 20 times. This low activity is in agreement with the observed fluxes and cannot be solely explained by relative protein quantification. This suggests that the activity of isocitrate lyase is partially tuned by acetylation *in vivo*.

All these observations suggest that the regulation of the isocitrate node involves two post-translational modifications, phosphorylation and acetylation, both acting at different levels: a gross regulation mechanism undertaken by AceK that blocks the flux through the TCA cycle and fine-tuning regulation of the glyoxylate shunt, which is partially inhibited by acetylation.

### Protein lysine acetylation regulates cellular motility and acid stress response

A high number of acetylated transcriptional regulators have been found in our study. The post-translationally modified transcriptional regulators identified in any condition represent 7% of the total acetylation sites found (Supplementary Table S3). Many of the acetylated peptides identified showed increased acetylation in the ΔcobB mutant. Since the acetylation of transcription factors may impair DNA-binding or protein–protein interactions, affecting transcriptional regulation, changes in protein acetylation in the ΔcobB knockout mutant may indirectly tune the transcription of many genes. To test this, DNA microarray studies were performed in *E. coli* wild-type and its knockout strain ΔcobB (Supplementary Tables S5, S6, S7 and S8; Supplementary Dataset S5). Genes differentially expressed in glucose exponential phase of batch cultures and steady state chemostat cultures in the ΔcobB mutant were analyzed using hierarchical clustering, showing that almost all significant changes were similar under both conditions. Genes related to bacterial motility were found to be most up-regulated in ΔcobB, while some genes involved in stress and pH response revealed down-regulation (Fig 5A).

The genes involved in the bacterial motility regulon can be classified into three functional groups according to the hierarchical regulation of their transcription (Fig 5B) (Liu & Matsumura, 1995; Claret & Hughes, 2002; Kalir & Alon, 2004). The up-regulation of class I, II and III genes suggest that probably one of the transcription factors regulating the *flhDC* operon is responsible for this global de-regulation. None of the transcriptional regulators of this operon showed differential gene expression in the ΔcobB mutant, suggesting that increased acetylation of a transcriptional regulator could be responsible for the differential expression of this regulon. The high alterations observed in both the expression of motility and chemotaxis genes and the acid resistance system in the ΔcobB mutant led

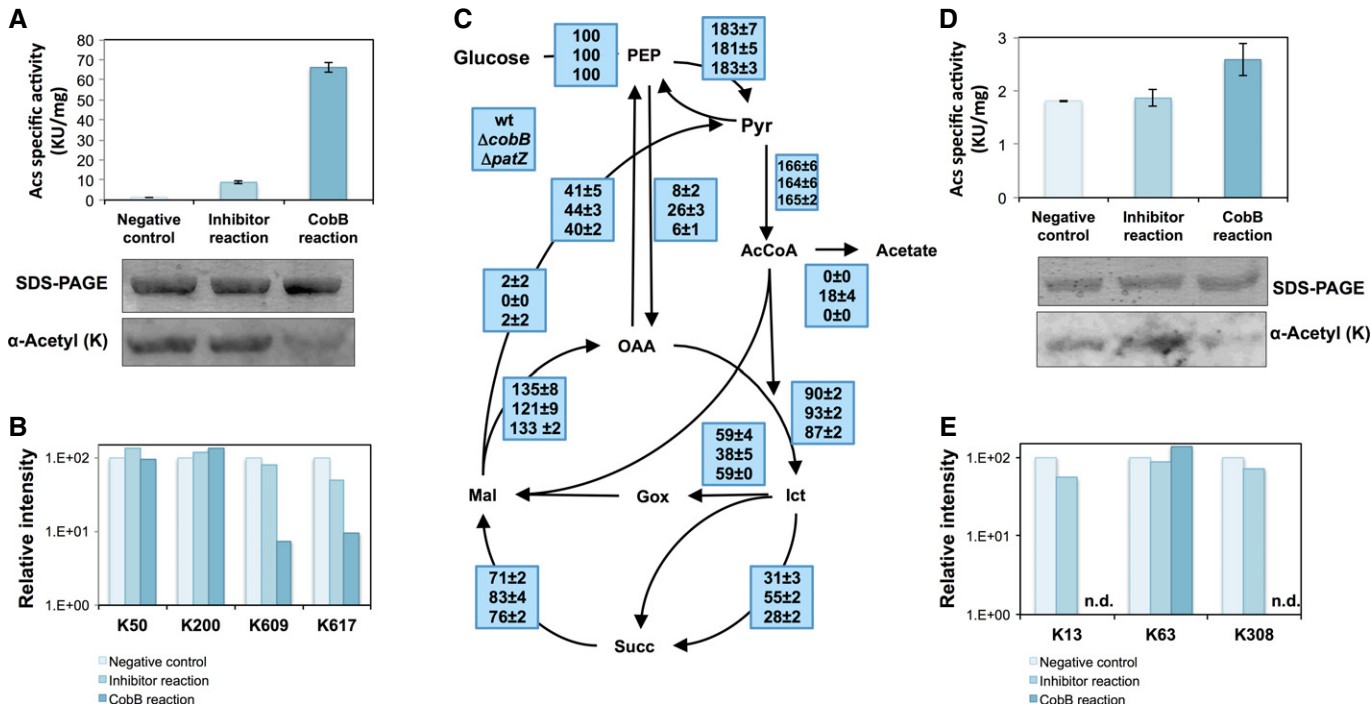

**Figure 4.    Regulation of acetate metabolism enzymes by lysine acetylation.**

A, B    *In vitro* deacetylation assays of acetyl-CoA synthetase (Acs). Affinity-purified enzymes were deacetylated with purified CobB. Negative controls were performed in the absence of CobB and in the presence of the CobB inhibitor nicotinamide (NAM). The effect of deacetylation was assessed by specific enzyme activity assays, Western blotting using an anti-acetyl-lysine antibody (A) and mass spectrometry (B). Relative intensities of the acetylated peptides found in each deacetylation reaction are shown for Acs in (B). Peptide intensities were normalized.

C    Metabolic fluxes in $^{13}$C-labeled glucose-limited chemostat cultures run at D = 0.2 h$^{-1}$. Fluxes are normalized to the glucose uptake rate (100%). Glucose uptake flux (mmol/g/h) was 8.66 ± 0.21 in the wild-type strain, 9.08 ± 0.06 in the ΔcobB mutant and 8.17 ± 0.01 in ΔpatZ mutant. Chemostat cultures were carried out in triplicate.

D, E    *In vitro* deacetylation assays of isocitrate lyase (AceA). Affinity-purified enzymes were deacetylated with purified CobB. Negative controls were performed in the absence of CobB and in the presence of the CobB inhibitor nicotinamide (NAM). The effect of deacetylation was assessed by specific enzyme activity assays, Western blotting using an anti-acetyl-lysine antibody (D) and mass spectrometry (E). Relative intensities of the acetylated peptides found in each deacetylation reaction are shown for AceA in (E). Peptide intensities were normalized.

Source data are available online for this figure.

us to think that the differential gene expression observed in the ΔcobB mutant could be caused by the inactivation of the transcription factor RcsB (Fig 5B and C).

It has been previously described that in *E. coli* K154 of RcsB is chemically acetylated by acetyl-phosphate and this lysine is deacetylated by CobB (Hu *et al*, 2013). This lysine is located in the helix-turn-helix (HTH) LuxR-type domain that directly interacts with its consensus DNA-binding sequence (Zhang *et al*, 2002). The acetylation of this residue could modify the interaction of RcsB with some of the promoters of the genes analyzed. In our proteomic results, we observed a higher acetylation of K154 of RcsB in the ΔcobB mutant. This fact, together with the differential gene expression of the flagella regulon and acid stress response genes in this mutant, indicates that the acetylation of this transcription factor impairs RcsB activity. To prove this hypothesis and observe the physiological consequences of the acetylation of this transcription factor, further experiments were carried out.

The acetylation of lysine 154 in the ΔcobB mutant increased the expression of the flagella regulon, provoking an increase in motility and number of flagella compared with the parent strain (Fig 6A1–2 and B1–2), showing a behavior similar to the *rcsB* null mutant

(Fig 6A3 and B3). Loss of function of acetylated RcsB also triggered the down-regulation of the acid stress response genes. Survival in an acidic environment requires glutamate, lysine or arginine decarboxylase enzymes. These proteins catalyze the decarboxylation of these amino acids consuming a proton in the reaction, thus maintaining the pH homeostasis. The genes belonging to glutamate-dependent acid response system (AR2), the acid response chaperones *hdeA* and *hdeB*, and others belonging to this survival system (Fig 5C), all of them activated by RcsB, were down-regulated in the ΔcobB mutant. Acid survival of *E. coli* strains was assessed after 2 h of incubation at low pH (2.5) (Fig 7). The deletion of *rcsB* impaired acid stress survival. In contrast, survival of the ΔcobB mutant, albeit highly affected, was slightly higher, which is consistent with low level expression of acid resistance-related genes in this mutant.

To demonstrate that the acetylation of lysine 154 is responsible for the phenotypic and transcriptional changes, site-directed mutagenesis was performed in order to mimic the electrostatic charge of a non-acetylated lysine (K154R) or an acetylated lysine (K154Q). Shifting this residue to a negatively charged glutamate (K154E) would mimic a succinylated lysine. These mutants were transformed into a

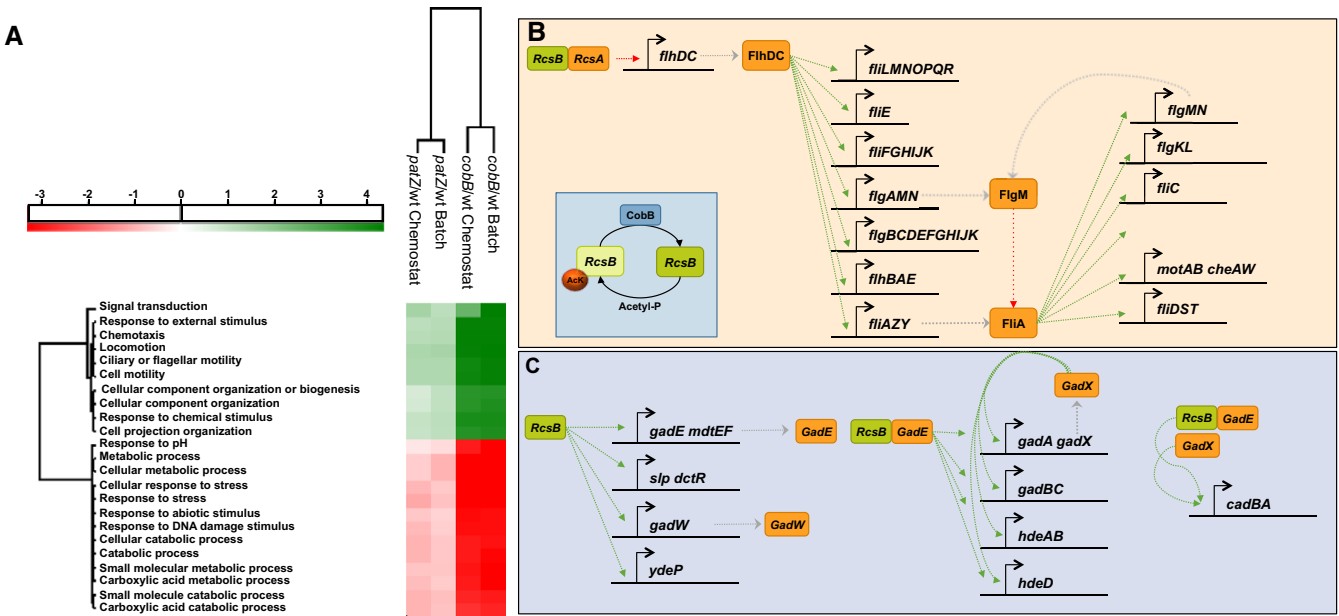

**Figure 5.  Microarray analysis of transcriptional response to *cobB* and *patZ* deletions in *E. coli* in batch and chemostat glucose cultures.**

A    Annotation matrix obtained from the gene expression microarray data where the main functions of the up- and down-regulated genes are grouped by their expression with a *P*-value threshold of 0.005; for further information see Geiger *et al* (2013). Gene expression data were referred to the wild-type strain and expressed as fold-change.

B, C    Transcriptional regulation of the flagellar regulon (B) and acid stress response genes (C) by RcsB (Keseler *et al*, 2013).

Source data are available online for this figure.

---

Δ*rcsB* background (Fig 6A4–6 and B4–6). The non-acetylated K154R RcsB mutant has a similar phenotype to the native construction, meaning that this mutation does not affect its function. In contrast, impaired electrostatic interaction with DNA in the K154Q and K154E RcsB mutations, mimicking the effect of a permanently acetylated or succinylated lysine, showed the same behavior as the Δ*cobB* mutant

(Fig 6A5–6 and B5–6). Similarly, the mutations of K154 of RcsB also affected acid stress survival. Acid survival was higher in the non-acetylated lysine mimic (K154R) and almost negligible in the K154Q and K154E acetylation and succinylation mimics (Fig 7B). Accordingly, glutamate decarboxylase activity was high in the wild-type, Δ*rcsB* complemented with *rcsB* and Δ*rcsB* containing the

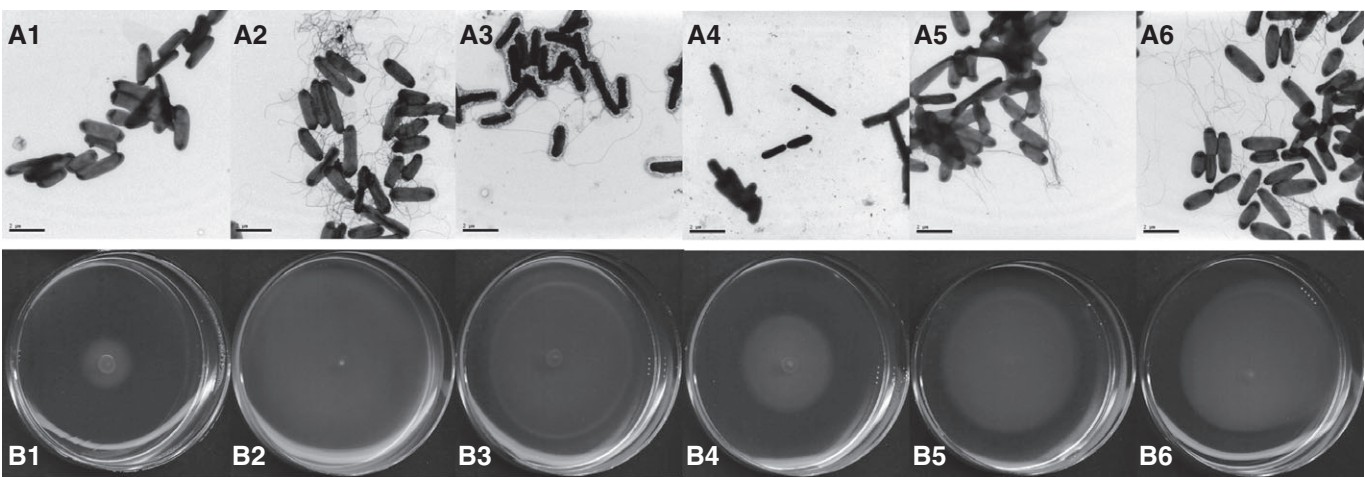

**Figure 6.  Physiological effects of the inactivation of RcsB due to the acetylation of lysine 154 in *E. coli*.**

A, B    Presence of flagella (A) and mobility assays in semisolid agar (B) of the *E. coli* wild-type strain (1) and mutants Δ*cobB* (2), Δ*rcsB* (3), Δ*rcsB*+p*rcsB*-K154R (4), Δ*rcsB*+p*rcsB*-K154Q (5) and Δ*rcsB*+p*rcsB*-K154E (6) was assessed. For both assays, *E. coli* wild-type and its mutants were harvested at the exponential phase of cultures.

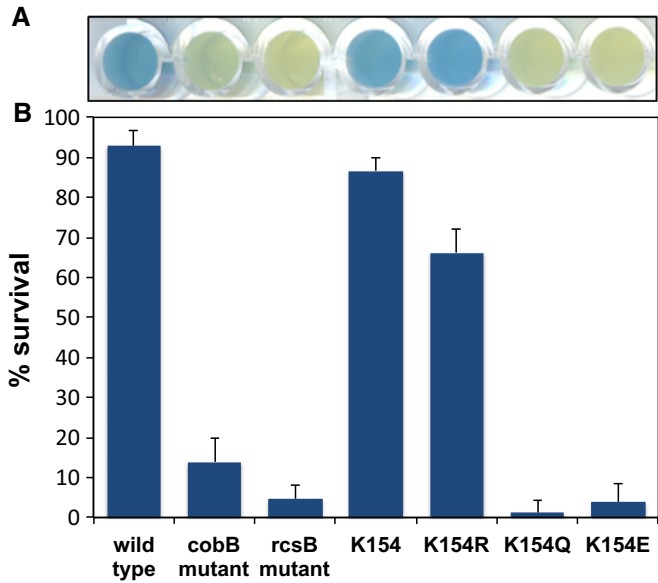

**Figure 7.   Effect of RcsB inactivation on acid stress response.**

A   Glutamate decarboxylase (Gad) enzyme activity. Colorimetric enzyme activity assay was followed by the consumption of H+ increasing the pH of the reaction *in vitro*. The increase of pH was detected by the color turn from yellow to blue of the pH indicator bromocresol green.

B   Acid stress survival of the different *E. coli* mutants. The effect of *cobB* and *rcsB* deletion on acid stress survival of *E. coli* was analyzed. The *rcsB* knockout mutant was complemented with the *rcsB* wild-type gene (K154) and its different mutants mimicking different acylation states of lysine 154 (K154R, K154Q and K154E). Bacteria were grown overnight in minimal medium with pH 5.5 to pre-adapt them to growth at low pH. These bacteria were subjected to acid stress for 2 h (pH 2.5), and cell survival was measured afterward. Mean values ± SD are shown (*n* = 3).

Source data are available online for this figure.

plasmid with the *rcsB* K154R mutant strains, while activity was low in ΔcobB and ΔrcsB mutants and in ΔrcsB complemented with *rcsB*-K154Q and K154E (Fig 7A).

Altogether, these data demonstrate that acetylation of lysine 154 of RcsB impairs its function, affecting flagella biosynthesis and bacterial motility and decreasing acid stress survival.

## Discussion

In recent years, several global acetylome characterizations have been reported in both prokaryotes and eukaryotes, but functional evidence for the diverse roles of reversible protein acetylation is still scarce. In this study, we addressed this issue by taking a systems biology approach, merging proteomic, transcriptomic, metabolomic and flux data with molecular biology studies. This allowed us to demonstrate that the deacetylase activity of CobB is global, contributing to the deacetylation of a large number of substrates. Therefore, deletion of CobB has a major impact on bacterial physiology. Lysine acetylation affects protein function and cell phenotype directly, modulating the activity of metabolic enzymes, or indirectly affecting transcriptional regulators. Regulation of particular physiological processes has been demonstrated *in vivo*: Protein deacetylation

by CobB activates acetate metabolism and regulates flagella biosynthesis, motility and acid stress survival.

The link between acetate metabolism and protein acetylation is well known (Starai *et al*, 2002; Gardner *et al*, 2006; Crosby *et al*, 2010; Weinert *et al*, 2013b; Kuhn *et al*, 2014). In addition to its well-demonstrated role in the regulation of acetyl-CoA synthetase, we have also shown that acetylation of isocitrate lyase contributes to regulating the flux through the glyoxylate shunt. Wang *et al* have recently described that protein acetylation affects the relative activity of glycolysis, gluconeogenesis and glyoxylate shunt in *S. enterica* by targeting the activity of the glyceraldehyde-phosphate dehydrogenase, isocitrate lyase and isocitrate dehydrogenase phosphatase/kinase (Wang *et al*, 2010). There is some controversy about the reproducibility of these results (Crosby *et al*, 2012a). In fact, we have not observed the acetylation of AceK in *E. coli*, and our results suggest that its activity is not regulated by acetylation.

Besides the existence of several protein acetyltransferases which may differ in their specificity toward targeted proteins and in the environmental signals to which they respond, there are increasing evidences that many proteins are either chemically or autocatalytically acetylated (Ramponi *et al*, 1975; Barak *et al*, 2004; Schwer *et al*, 2009; Kuo & Andrews, 2013; Weinert *et al*, 2013a,b, 2014; Kuhn *et al*, 2014). The high protein acetylation ratios of the ΔpatZ mutant in acetate cultures are in agreement with this hypothesis and suggest that PatZ could modulate chemical acetylation by regulating the levels of acetylating metabolites. A high acetyl-CoA synthetase activity in the *patZ* mutant could lead to increased intracellular concentration of acetyl-CoA. However, deletion of *patZ* did not alter the pool of acetyl-CoA, succinyl-CoA and free CoA (Supplementary Fig S10). Acetyl-phosphate could also be responsible for this acetylation pattern; in fact, the activity of phosphotransacetylase (Pta) also increased in the ΔpatZ mutant (Supplementary Fig S6). It could be argued that, during evolution, organisms may have evolved two complementary strategies in order to fight chemical acetylation: acetyltransferases, regulating the synthesis of acetylating agents, and deacetylases, removing acyl moieties from proteins.

The abundance of acid residues in the protein acetylation motif can shed light on the specificity of protein modification and explain why chemical acetylation is frequent even under mild physiological conditions. Acid residues are also over-represented in the lysine acetylation motifs of mitochondrial and cytoplasmic proteins in rat (Lundby *et al*, 2012) and other microorganisms (Okanishi *et al*, 2013). Aspartate and glutamate residues in the vicinities of targeted lysines could enhance the nucleophilicity of lysine residues, which could attack acetylating agents. This mechanism would be in agreement with both enzymatic and chemical acetylation (Yan *et al*, 2002; Smith & Denu, 2009; Weinert *et al*, 2013b).

We have demonstrated that increased acetylation of RcsB exerts several physiological effects: It impairs flagella synthesis, motility and chemotaxis and compromises acid stress survival. Switching the electrostatic charge of the residue 154 of RcsB demonstrated that the positively charged side chain of K154 is essential for DNA binding, since K154Q, K154R and K154E mutants showed similar phenotypes. This suggests that any post-translational modification affecting local charge at this residue, including other acylations such as succinylation (Colak *et al*, 2013), would yield a similar effect. Interestingly, the transcriptomic effects shown in the ΔcobB mutant were similar to those observed in ΔpatZ, probably due to

increased chemical acetylation in this mutant (Fig 5). Recently, acetylation of K154 of RcsB has been described, although these authors claimed that it did not affect motility, probably due to the use of a different strain for the motility tests. However, we also observed that increased acetylation of RcsB in K154 represses transcription of *rprA* gene as described in the same study (Hu *et al*, 2013).

The phenotype of the Δ*cobB* mutant was significantly affected, mirroring the peptide acetylation ratios of proteins, presumably, due to the acetylation of proteins at lysine residues which are key for their activity (Fig 2C; Supplementary Table S2). However, this was not the case for the Δ*patZ* mutant, whose growth was not affected in acetate cultures despite the high peptide acetylation ratios observed. An unaltered phenotype despite a severely affected acetylation pattern was also described before in an acetate kinase (*ackA*) mutant (Weinert *et al*, 2013b; Kuhn *et al*, 2014). This shows that not all acetylation events have an effect on protein function, and an increase in acetylation of proteins may have no evident effect on cell physiology.

Despite the widespread acetylation of proteins, the number of known substrates of the sirtuin CobB is limited. The discovery of new substrates of CobB has been driven by *in vitro* techniques (Starai *et al*, 2002; Thao *et al*, 2010; Hu *et al*, 2013; Zhang *et al*, 2013) and also high-resolution MS-based proteomics. In a previous study (Weinert *et al*, 2013a), the number of CobB substrates ($\log_2$ acetylation ratio > 1 for Δ*cobB* mutant compared to wild-type) was approximately 10% (≈366 peptides), while in our study 40% of all the acetylated peptides detected are over this cutoff value in at least one condition (1,025 peptides). Interestingly, both datasets reveal that there is no relevant deacetylation motif for CobB, which preferentially recognizes acetyl-lysine residues in disorganized regions of the proteins or in the protein termini (Khan & Lewis, 2005; Weinert *et al*, 2013b). The high chemical reactivity of the lysine side chain may indicate that neutralization of its positive charge by acetylation could alter protein function. A large resource of targeted lysine residues has been built in this study, containing potential wealth information on the physiological roles of protein acetylation. These data will need further validation. We have here targeted roles of the differential acetylation of three proteins, although several acetylation events can be understood based on the bibliography. This is the case of K326 of the gluconeogenic fructose-1,6-bisphosphate aldolase class II (FbaA), whose mutation leads to a loss of 94% of activity (Zgiby *et al*, 2000), and K119 of molybdopterin synthase (MoaE), whose mutation inhibits its activity completely (Wuebbens & Rajagopalan, 2003). Still many functions of lysine acetylation in bacteria are poorly understood, and efforts are needed to complete the picture of the regulatory roles of post-translational acetylation of proteins in bacterial physiology, to which the here described data can make an important contribution.

# Materials and Methods

### *Escherichia coli* strains and culture conditions

*Escherichia coli* wild-type BW25113 and its knockout strains (Supplementary Table S9) were grown in minimal media in batch mode with glucose and acetate as described in Castaño-Cerezo *et al*

(2011) and in glucose-limited chemostat at a dilution rate of 0.2 h (Nanchen *et al*, 2006; Renilla *et al*, 2012).

## Proteomics

### *Sample preparation for lysine acetylation mapping*
Cells were harvested at exponential and stationary phase in glucose cultures, exponential phase in acetate cultures and in steady state in glucose-limited chemostats. Cell pellets were washed three times with PBS and then resuspended in lysis buffer containing 8 M urea, 50 mM ammonium bicarbonate and 1 tablet of complete mini EDTA-free cocktail (Roche, Boehringer Mannheim) and supplemented with 10 mM nicotinamide and 10 μM trichostatin in order to inhibit deacetylases. Cells were sonicated on ice for three cycles (20 s each) with a probe of 3 mm of diameter in a Vibra Cell VC 375 ultrasonic processor (Sonics Materials, Danbury, CT). The lysate was clarified by centrifugation for 20 min at 20,000 × *g* at 4°C.

Three mg of protein of each condition and strain were reduced with 2 mM dithiothreitol for 30 min at 56°C and alkylated with 4 mM iodoacetamide during 20 min in the dark, followed by LysC (1:75) digestion during 4 h at 37°C. Samples were diluted fourfold in 50 mM ammonium bicarbonate buffer and digested with trypsin (Promega, Madison, WI) (1:100) during 16 h at 37°C.

For quantitative analysis of peptide lysine acetylation, stable isotope dimethyl labeling was used as described in Boersema *et al* (2009). Labeled peptides from each strain were mixed in a 1:1:1 proportion. Acetylated peptides were immunoprecipitated as described (Choudhary *et al*, 2009). Nine milligrams of peptides was resuspended in immunoprecipitation buffer (50 mM MOPS, 10 mM sodium phosphate and 50 mM NaCl pH 7.4). The peptide solution was mixed with 100 μl of anti-acetyl-lysine antibody beads (ImmuneChem, Burnaby, Canada) and incubated for 16 h at 4°C. Beads were washed four times with immunoprecipitation buffer, twice with water and eluted with 0.1% trifluoroacetic acid. Lysine-acetylated peptides were desalted using $C_{18}$ StageTips.

### MS specifications

Samples were resuspended in 10% formic acid (FA)/5% DMSO, and 40% of the sample was analyzed using a Proxeon Easy-nLC100 (Thermo Scientific) connected to an Orbitrap Q-Exactive mass spectrometer. Samples were first trapped (Dr Maisch Reprosil C18, 3 μm, 2 cm × 100 μm) before being separated on an analytical column (Agilent Zorbax SB-C18, 1.8 μm, 40 cm × 75 μm), using a gradient of 60 min at a column flow of 150 nl/min. Trapping was performed at 8 μl/min for 10 min in solvent A (0.1 M acetic acid in water), and the gradient was as follows: 7–30% solvent B (0.1 M acetic acid in acetonitrile) in 91 min, 30–100% in 3 min, 100% solvent B for 2 min and 7% solvent A for 18 min. Nanospray was performed at 1.7 kV using a fused silica capillary that was pulled in-house and coated with gold (o.d. 360 μm; i.d. 20 μm; tip i.d. 10 μm). The mass spectrometers were used in a data-dependent mode, which automatically switched between MS and MS/MS. Full-scan MS spectra from *m/z* 350 to 1,500 were acquired at a resolution of 35,000 at *m/z* 400 after the accumulation to a target value of $3 \times 10^6$. Up to ten most intense precursor ions were selected for fragmentation. HCD

fragmentation was performed at normalized collision energy of 25% after the accumulation to a target value of $5 \times 10^4$. MS2 was acquired at a resolution of 17,500, and dynamic exclusion was enabled (exclusion size list 500, exclusion duration 30 s).

## Data analysis

Raw data were analyzed by MaxQuant (version 1.3.0.5) (Cox & Mann, 2008). Andromeda (Cox *et al*, 2011) was used to search the MS/MS data against the Uniprot *E. coli* MG1655 database (version v2012-09, 4,431 sequences), including a list of common contaminants and concatenated with the reversed version of all sequences. Trypsin/P was chosen as cleavage specificity allowing three missed cleavages. Carbamidomethylation (C) was set as a fixed modification, while Oxidation (M), Acetyl (Protein N-term) and Acetyl (K) were used as variable modifications. For dimethyl labeling, Dimethyl-Lys0 and DimethylNter0 were set as light labels, DimethylLys4 and DimethylNter4 were set as medium labels, and DimethylLys8 and DimethylNter8 were set as heavy labels. All peptides were used for quantification studies, but to find significantly acetylated peptides (Fig 2), only those found in two or more replicates and with a FDR < 0.05 were used (*t*-test adjusted for multiple testing using permutation-based FDR). The database searches were performed using a peptide tolerance of 20 ppm for the first search and 6 ppm for the main search. HCD fragment ion tolerance was set to 20 ppm. Data filtering was carried out using the following parameters: Peptide false discovery rate (FDR) was set to 1%; Andromeda score was set to 30; max peptide PEP was set to 1; minimum peptide length was set to 5; minimum razor peptides were set to 1; peptides used for protein quantification was set to razor and unique peptides; protein quantification was performed by using only unmodified peptides and Oxidation (M) and Acetyl (Protein N-term); and the re-quantify option was enabled. Further data processing was performed using the Perseus tool (version 1.3.0.4) available in the MaxQuant environment.

Lysine acetylation motif was created using the IceLogo software package. For the generation of the acetylation motif, all acetylated peptides with an Acetyl (K) Probability higher than 0.9 and a cutoff *P* < 0.01 were used.

## In vitro enzyme activities

Acetyl-CoA synthetase (Acs), isocitrate dehydrogenase (Icd), isocitrate lyase (AceA) and phosphotransacetylase (Pta) were assayed as previously described (Castaño-Cerezo *et al*, 2009) and acetate kinase as described by Foster and collaborators with minor modifications (Foster *et al*, 1974).

## DNA microarray

Global gene expression was assessed in glucose exponential phase and chemostat cultures. RNA was purified using Vantage RNA purification kit (Origene, MD, USA). Purity and concentration of isolated RNA were assessed in a NanoDrop ND-1000 spectrophotometer (NanoDrop Technologies, Wilmington DE). Quality was evaluated by microfluidic capillary electrophoresis on an Agilent 2100 Bioanalyzer (Agilent Technologies, Palo Alto, CA) using Agilent RNA 6000 Pico kit. GeneChip *E. coli* Genome 2.0 arrays (Affymetrix, Santa Clara, CA) were prepared and loaded according to the manufacturer's instructions. Signal extraction and normalization was performed using GeneChip Expression Console, and RMA algorithm was applied (Smyth, 2004). Log$_2$ signals were loaded into Babelomics, and Class Comparison analysis were performed using Limma method (FDR 0.05) (Medina *et al*, 2010). Hierarchical clustering and matrix annotation were performed using Perseus (Cox & Mann, 2008).

## Protein purification and *in vitro* deacetylation assays

### Protein purification

The acetyl-CoA synthetase (Acs), isocitrate lyase (AceA) and the NAD$^+$-dependent deacetylase (sirtuin) CobB from *E. coli* BW25113 were expressed using ASKA clone plasmids (GFP$^-$) (Kitagawa *et al*, 2005). In order to obtain hyper-acetylated proteins, plasmids were transformed into *E. coli* BL21 Δ*cobB* or K12 Δ*cobB* for protein expression. The transformants were grown for 14 h at 28°C with 0.1 mM of IPTG induction. Cells were harvested by centrifugation and washed three times with 0.9% NaCl and 10 mM MgSO$_4$. Cell pellets were resuspended in binding buffer (15.5 mM Na$_2$HPO$_4$, 4.5 mM NaH$_2$PO$_4$, 500 mM NaCl and 20 mM imidazole, pH 7.4) and lysed by sonication (3 × 30″ cycles) on ice. Cell debris was removed by centrifugation, and protein extract was loaded onto His GraviTrap columns (GE healthcare, Buckinghamshire, UK). His-tagged proteins were purified according to the manufacturer protocol. Purified proteins containing imidazole were cleaned using Amicon ultra 4 centrifugal filters (Millipore, Country Cork, Ireland).

### In vitro *protein deacetylation assays*

For deacetylation assays, 100 μg of protein (Acs or AceA) was incubated with CobB (5 μg for Acs and 50 μg for AceA deacetylation assays, respectively) in deacetylation buffer (50 mM HEPES, 5 mM NAD$^+$, 1 mM DTT and 5% glycerol, pH 7.0) during 1 h at 37°C with NAD$^+$ (5 mM) as substrate.

After incubation, aliquots of the reaction were used for enzyme activity assays, Western blotting or mass spectrometry for lysine acetylation mapping (Supplementary Materials and Methods).

## Metabolic flux ratio analysis and $^{13}$C-constrained metabolic flux analysis

Metabolic flux ratio analysis in *E. coli* BW25113 and its knockout mutants Δ*cobB*, Δ*patZ*, Δ*cobB*Δ*aceK* and Δ*patZ*Δ*aceK* in glucose batch exponential phase and chemostat cultures (D = 0.2 h$^{-1}$) was performed as previously described (Zamboni *et al*, 2009). One milligram of cells was washed twice with 1 ml of 0.9% NaCl and 10 mM MgSO$_4$ and hydrolyzed in 300 μl 6 M HCl at 105°C for 15 h in sealed 1.5-ml tubes. The hydrolysates were dried in a heating block at 85°C under a stream of air and then derivatized at 85°C for 60 min in 30 μl of dimethylformamide and 30 μl of N-(tert-butyldimethylsilyl)-N-methyl-trifluoroacetamide with 1% (v/v) tert-butyldimethylchlorosilane with slight shaking (Zamboni *et al*, 2009). One microliter of the derivatized sample was injected into a 6,890 N Network GC system, combined with a 5,975 Inert XL Mass Selective Detector (Agilent Technologies). The gas chromatography–mass spectrometry-derived mass isotope distributions of proteinogenic amino acids

were corrected for naturally occurring isotopes (Fischer & Sauer, 2003), and nine ratios of fluxes through converging reactions were determined. Calculations were performed using the Matlab-based software FiatFlux 1.66 (Zamboni *et al*, 2005).

Intracellular net carbon fluxes were estimated by using the stoichiometric model previously described (Zamboni *et al*, 2009) that included all major pathways of central carbon metabolism, including the glyoxylate shunt and the Entner–Doudoroff (ED) pathway. The matrix consisted of 25 reactions and 21 metabolites. Net fluxes were then calculated using: (i) the stoichiometric reaction matrix, (ii) the relative metabolic flux ratios, (iii) physiological data, and (iv) precursor requirements for biomass synthesis. Specifically, the following flux ratios were used: serine derived through the EMP pathway, pyruvate derived through the ED pathway, oxaloacetate (OAA) originating from PEP, PEP originating from OAA, OAA originating from glyoxylate, the lower and upper bounds of pyruvate originating from malate and the upper bound of PEP derived through the PP pathway.

### Molecular biology

The *rcsB* and *cobB* genes were PCR-amplified from *E. coli* BW25113 genomic DNA and cloned into the *pBAD24* plasmid (Guzman *et al*, 1995). Single amino acid mutants K153R, K153Q and K153E (for *rcsB* gene) and H110Y (for *cobB* gene) were obtained by site-directed mutagenesis. The Stratagene kit for mutagenesis was used according to the manufacturer instructions. Mutagenesis primers are listed in Supplementary Table S10.

### Electron microscopy for flagella observation

*Escherichia coli* strains were grown in glycerol minimal medium. The cells were harvested at mid-exponential phase and fixed with 3% glutaraldehyde. After three washes with PBS, each cell suspension was placed on electron microscopy grids and stained for 15 s with 2% uranyl acetate before flagella examination in a JEM-1011 Electron Microscope (Jeol, Tokyo, Japan) operating at 90 kV.

### Mobility assays

All strains were grown until mid-exponential phase. Five microliter of these cultures was inoculated into semisolid agar (10 g/l tryptone, 5 g/l NaCl and 0.25% agar). Plates were checked for mobility after 16-h incubation at 30°C.

### Acid stress resistance test

All tested strains were grown overnight in glycerol minimal medium (pH 5.5) supplemented with 1.5 mM glutamic acid. A 1:1,000 dilution of the overnight culture was inoculated in glycerol minimal medium pH 2.5 supplemented with 1.5 mM glutamic acid. Cell survival was measured after 2 h in acidic media as previously described (Krin *et al*, 2010).

### Glutamate decarboxylase activity (GAD)

For Gad enzyme activity determination, approximately $3 \times 10^8$ stationary-phase cells grown in glycerol minimal medium at pH 5.5

were harvested by centrifugation (4,000 *g*, 4°C) and washed twice with 0.9% NaCl. Cells were mixed one to one with the Gad reagent (1 g/l L-glutamic acid, 0.05 g/l bromocresol green, 90 g/l NaCl, 0.3% (v/v) Triton X-100) and incubated for 30 min at 35°C. The presence of this enzyme was monitored by the color shift of bromocresol green, which turns from yellow to blue (Deininger *et al*, 2011).

### Data availability

The mass spectrometry proteomics data have been deposited to the ProteomeXchange Consortium (http://proteomecentral.proteomexchange.org) via the PRIDE partner repository (Vizcaíno *et al*, 2013) with the dataset identifier PXD001226.

The transcriptomic data discussed in this work have been deposited in NCBI's Gene Expression Omnibus (Edgar *et al*, 2002) and are accessible through GEO Series accession number GSE62094 (http://www.ncbi.nlm.nih.gov/geo/query/ acc.cgi?acc = GSE62094).

**Supplementary information** for this article is available online: http://msb.embopress.org

### Acknowledgements

We wish to thank J. Pozueta-Romero and José Maria Pastor for fruitful discussions. A. Torrecillas (CAID, University of Murcia) is gratefully acknowledged for his help in proteomic analysis. J. Cantalejo (Parque Científico de Madrid) is acknowledged for microarray analysis. The PRIDE and GEO teams are acknowledged for their support for the publication of raw proteomic and transcriptomic data. S. Castaño-Cerezo is a recipient of a PhD fellowship from Fundación Séneca (CARM, Murcia) and an EMBO short-term fellowship. V. Bernal acknowledges a post-doctoral contract from Universidad de Murcia (Programa Propio). This work has been partly funded within the framework of PRIME-XS, Grant Number 262067, funded by the European Union 7th Framework Program, and the MICINN BIO2011-29233-C02-01 and Fundación Séneca-CARM 08660/PI/08 projects. SCC, HP, SC, AJRH and AFMA acknowledge additional support from the Netherlands Proteomics Centre.

### Author contributions

SCC and VB formulated the original idea of the project. SCC, VB, TF, US, AFMA and MC designed the experiments. SCC, VB, HP, TF, NCSD and AFMA performed the experiments. SCC, VB, SC, and TF analyzed the data. SCC and VB drafted the manuscript. SCC, VB, TF, US, AJRH, AFMA, MA and MC contributed to the study design, discussed the results and commented on the manuscript. All authors read and approved the final version of the manuscript.

### Conflict of interest

The authors declare that they have no conflict of interest.

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
