## [Review Process File · Molecular Systems Biology]

Protein deacetylation affects acetate metabolism, motility and acid stress response in *Escherichia coli*

Sara Castano-Cerezo, Vicente Bernal, Harm Post, Tobias Fuhrer, Salvatore Cappadona, Nerea C. Sánchez-Díaz, Uwe Sauer, Albert J.R. Heck, A.F. Maarten Altelaar, Manuel Cánovas

Corresponding author: Vicente Bernal, University of Murcia

Review timeline:

Submission date:	06 March 2014
Editorial Decision:	15 April 2014
Revision received:	07 August 2014
Editorial Decision:	26 September 2014
Revision received:	14 October 2014
Accepted:	23 October 2014

Editor: Maria Polychronidou

Transaction Report:

1st Editorial Decision

15 April 2014

Thank you again for submitting your work to Molecular Systems Biology. We have now heard back from the three referees who agreed to evaluate your manuscript. As you will see from the reports below, the referees acknowledge that you address a potentially interesting topic. However, they raise a series of concerns, which should be carefully addressed in a revision of the manuscript.

Overall, the reviewers refer to the need to perform additional experiments in order to better support the main conclusions. Without repeating all the comments listed below, among the more fundamental points are the following:

- Further experimentation is required to demonstrate that the effects of CobB on bacterial physiology are mediated by its deacetylase activity and that CobB regulates isocitrate lyase via deacetylation.
- Additional analyses providing further insights into the unexpected finding that loss of PatZ results in elevated acetylation levels in acetate media would significantly enhance the impact of the study. Referee #2 provides constructive comments regarding this point.

On a more editorial level, we would like to ask you to deposit all large-scale datasets in one of the relevant public databases and to include the accession codes in the Materials & Methods section.

Reviewer #1:

This is an interesting manuscript, that gives new insights into the role of protein acetylation in *E. coli*. Recently, there has been a great deal of discussion about the enzymatic-vs non-enzymatic lysine-acetylation and the question whether non-enzymatic acetylation still has a function. This manuscript

clearly shows cellular functions of lysine-acetylation, validated by site-directed mutagenesis. It goes well beyond previous lysine-acetylome publications in its completeness and follow-up experiments.

The figures supporting the results are unfortunately of a low quality. The fonts are too small everywhere. Combination of red-green should be avoided to help colorblind scientists. There is no mention of which sequences were selected to produce figure 2A.

There is no explanation of colors in figure 2B. Figure 3 has sub-panels A, B, C, E, F (no D) and no caption for F. The metabolic fluxes don't have unit of measurement. How many chemostat cultures were run, is also missing. Figure 5, no explanation of kind of assays is given. Figure 6, no explanation of colors in colorimetric assays, no labels.

Thus my recommendation to majorly revise the figures before a resubmission.

Reviewer #2:

Castano-Cerezo, Canovas, and colleagues applied a combination of proteomic, transcriptomic, and metabolic flux analyses to analyze the roles of the sirtuin deacetylase CobB and the protein acetyltransferase PatZ in *E. coli* grown under several conditions. Based on these analyses they find that CobB is important for growth in glucose-limiting chemostat cultures and in acetate-media, notably, the effect of CobB-deficiency on acetylation levels under these same conditions is markedly greater than in glucose-batch culture. Surprisingly, they find that PatZ deficiency caused globally elevated acetylation in acetate-media. Metabolic flux analysis showed that CobB-deficiency reduced flux through the glyoxylate shunt and increased acetate production. Transcriptome analysis revealed that CobB-deficiency affected the transcription of genes involved in bacterial motility and acid stress resistance. Additional experiments provided mechanistic support for the observed metabolic flux and transcriptome changes by showing that CobB regulates acetyl-CoA synthetase, isocitrate lyase, and the transcription factor RcsB.

This study combines several systems-level analyses to characterize wild-type, CobB-deficient, and PatZ-deficient bacteria. Their experiments appear well performed and the data are appropriately interpreted, and authors candidly discuss different interpretations of their results. However, acetylation sites analyzed on Acs and RcsB has been partially characterized previously, and at this stage, the novelty of their results is not very clear. We suggest additional experiments that would add further insights into how CobB deletion affects *E. coli* physiology, and together with their global investigations, these additional results would make the manuscript a strong candidate for publication in *Molecular Systems Biology*.

Major Concerns

1. Deletion of *patZ* caused an increase in global acetylation (Fig. 1D) in *E. coli* cultured in acetate containing media. This increase is at least as prominent, if not more, as seen in CobB deletion mutants cultured in different conditions (Fig. 1E-H). This suggests that in acetate cultures, *patZ* suppresses global acetylation to a greater degree than CobB. Strikingly, however, the authors do not observe a strong growth phenotype in *PatZ* mutants under these conditions, and they clearly acknowledge this as "Despite the high impact of *patZ* deletion on protein acetylation, no evident phenotypic effects were observed." This indicates that the global increase in acetylation was not sufficient to explain the phenotype of the CobB strain. It raises questions as to whether CobB regulates acetylation of a different set of sites than *PatZ* to regulate biological functions, and whether all phenotypic effects observed in CobB are related to its deacetylase activity? Currently, the authors conclude that the phenotypes in CobB are related to its deacetylase function; however, it has not been demonstrated directly. To address point, the authors should generate a mutant strain expressing a deacetylase inactive mutant of CobB, and compare its effects on global acetylation and *E. coli* phenotype to CobB deletion mutants. These experiments will address whether CobB affects bacterial physiology through its deacetylase activity, or whether CobB has deacetylase independent functions in cells.

Additionally, it would be interesting to know how CobB targets were affected in *PatZ* deficient cells grown in acetate containing media.

2. The authors perform functional analysis of acetylated proteins (Page 6, Figure 2B) and conclude that acetylation occurs more frequently on metabolic proteins, and that metabolic proteins are often multiply acetylated. However, proteome-wide studies of acetylation are biased towards abundant

proteins and the frequency of observed acetylation is correlated to protein abundance. Many proteomic studies of acetylation have made the mistake of misinterpreting the technical bias to detect acetylation on highly abundant metabolic proteins to indicate that metabolic proteins are more highly acetylated than other proteins, the authors should avoid making the same mistake. The results shown in Figure 2B may simply reflect the abundance of these proteins. It would be more appropriate to correct acetylation abundance bias to protein abundance, and repeat these analyses. At a minimum the authors should indicate that these results could be attributed to the abundance of metabolic proteins or the analysis should be removed altogether.

3. Their more novel finding, that CobB regulates isocitrate lyase through deacetylation, is not supported well-enough. The authors find that isocitrate lyase (Icd) protein abundance was slightly reduced (~50%) in CobB-deficient bacteria, however, Icd activity was ~20-fold reduced in cell free extracts from CobB-deficient cells grown in chemostat culture. The authors conclude that acetylation fine-tunes Icd activity. However, the authors were unable to detect altered Icd activity after treating purified Icd with CobB, which they explain may be due to the purified enzyme being present in an active, non-acetylated form. In order to provide better experimental evidence that CobB regulates Icd through direct deacetylation, rather than by affecting its protein levels or through an unknown mechanism, the authors should purify Icd under conditions where its activity is affected, presumably, by acetylation and then treat the purified protein with CobB to test whether this treatment rescues Icd activity. For example, using Icd purified from CobB-deficient cells grown in chemostat culture.

The panels in Figure 3 are not labeled correctly (E and F, versus D and E in the text and figure legend). In addition the text states that "isocitrate lyase and isocitrate dehydrogenase levels were measured (Fig. 3D and 3E)" while the figure legend indicates "Enzyme activities of (B)Acs, (D)AceA, and (E) Icd..." the gene symbols should be defined in either the text or the figure legend, or both. This could be made even clearer by indicating the enzyme activity being measured on the axis of the column chart shown in the figure itself.

4. The authors make the interesting finding that acetylation levels are elevated in PatZ-deficient bacteria grown in acetate media. In the discussion they present a reasonable explanation that this may be due to elevated activity of Acs in PatZ-deficient cells, resulting in elevated acetyl-CoA levels and therefore globally elevated acetylation. The manuscript would be strengthened by performing some fairly straightforward experiments to test this theory. The authors could assay Acs activity (which they have done already in this study) in PatZ-deficient cells grown in acetate media, compared with other growth conditions and wild-type cells. They could further compare acetyl-CoA levels between these strains and growth conditions. These experiments would strengthen the manuscript by providing further insight into the only remarkable finding attributed to the loss of PatZ.

Minor concerns

1. The reviewing a manuscript could be made much easier by providing both page numbers, line numbers, and labeling figures with the figure number.
2. Acetylation has been known to occur since the 1960s (the authors indicate 1970s)
3. The iceLogo comparison is not sufficiently explained in the methods section. The authors should indicate what the acetylated sequences were compared to?
4. All MS results should be provided in Excel spreadsheets for easier navigation of the results.
5. Fig. S3B, the authors note a decreased acetylation of K609 on Acs in PatZ cells. However, acetylation also appears somewhat reduced in CobB, which appear inconsistent with this site being deacetylated by CobB.
6. The authors indicate "Flux through the glyoxylate shunt decreased by 50% in the....." However, the Fig. 3C, seems to indicate that the flux decrease was ~33%?
7. Fig. S5 appears redundant with Fig. 4A, as the Fig. S5 does not provide information about the clustered terms.
8. Fig. 6, the authors should clearly explain how to interpret the provided images. The readership of MSB is not necessarily familiar with these types of analyses and some explanation would help them better understand the provided data.

Reviewer #3:

The study by Castano-Cerezo et al. ('Protein deacetylation affects acetate metabolism, motility and acid stress response in *Escherichia coli*') utilizes a SILAC-based proteomics approach to decipher targets of lysine acetylation / deacetylation in CobB (a prokaryotic sirtuin-like lysine deacetylase) and lysine acetyltransferase (PatZ) mutants grown in chemostat batch cultures. The authors further explore the relationship between lysine acetylation / deacetylation and 3 phenotypes - acetate metabolism, flagellar motility and acid stress response.

The manuscript represents a significant body of analytical work, which has been performed competently and the manuscript is well presented. However, for publication in MSB, there needs to be a major advancement that I am unable to find in the current work. The authors themselves quote the work of Wang et al (Science 2010) and Weinert et al (Mol. Cell 2013). The current work outperforms Wang in terms of sheer numbers, but similar identifications and numbers were achieved by both Weinert and in a more recent study by Colak et al (MCP 2013), which is not (but should be) referenced here. All 3 studies make the link between CobB and metabolism. In several places, the study is largely observational, with real hypotheses remaining untested. One good example is in the section on deficient function of the glyoxylate shunt and a link with isocitrate lyase - using acetylation mimics and recombinant proteins for functional assays could provide a functional link between this enzyme and the phenotype and the study would be greatly enhanced if such an approach was undertaken. The authors have got part of the way there by showing a decrease in isocitrate lyase function in the absence of cobB but without linking that to a specific cobB-regulated acetylated lysine, the hypothesis remains unproven.

The potentially really interesting findings of the current study are those related to motility and acid stress, as this reviewer is unaware of previous results showing such a relationship. Here however, it would be crucial to examine CobB complements, to determine if restoration of the gene also restores the WT phenotype. Additional experiments and comments are below. It would also be useful to determine if there are off-target effects.

1. For the mapping experiments of lysine acetylated peptides, the authors state that 4 biological replicates were used, but do not state the basis for how the data from each replicate were included. Was a peptide included if it was identified once in one biological replicate? What confidence thresholds were applied (scoring parameters etc.)? How were site allocations defined? In the quantification studies, were peptides only confidently quantified if they were identified in 1,2,3 or 4 experiments, and what statistical tests were performed? I think in summary, substantially more information may be required here.
2. Results, p.2, 'Deletion of cobB caused substantial phenotypic changes' and then all following experiments for example the flux experiments etc. - these studies would all be strengthened by the inclusion of a cobB complement (as above).
3. When using the anti-lysine acetylation antibody and performing quantitation, were the unbound proteins / peptides also analysed to control for non-specific binding and loss of acetylated proteins? It is generally useful to determine whether whole protein abundances change under chosen conditions. The authors use transcript arrays to examine the genomic response to cobB deletion but they most likely also have access to the proteome-level data. These should be included.
4. Results, p.1 '...in exponential phase the patZ gene expression is low...' Did the authors confirm this or is this previously published? Also '...might be explained by the presence of at least 25 putative acetyltransferases...' - another reason to examine unmodified protein abundances to determine whether the abundance of any of these is altered by patZ deletion.
5. Results, p.2, 2nd para - if CobB is the major deacetylase, what is the explanation for the (relatively) low proportion of acetylated peptides influenced by cobB deletion (17%-up to 30% in acetate)? Furthermore, in the Results on p.3, para 3, the authors state that 8 acetylation sites on Acs were unaffected by cobB deletion. So in some respects the results indicate that CobB is not the major deacetylase, but perhaps one of two or more deacetylases?
6. Results, 3rd para - a motif analysis of CobB substrates is presented in Weinert et al.
7. How specific in *E. coli* is nicotinamide to CobB? Nicotinamide in eukaryotes has substantial off-target effects other than inhibition of Sirts, for example, on NAD-dependent enzymes. This should be addressed.
8. As an additional control, does nicotinamide (or other sirt/CobB inhibitor) influence motility and /

or the expression of flagellar genes?

9. The studies specifically examining the lysine 154 residue in RcsB are a highlight of the study, but something doesn't add up. While clearly loss of CobB increases flagellar biosynthesis and motility (A/B1-2), mimicking the non-acetylated lysine appears to alter the cell morphology, and the motility is increased relative to wild-type. This suggests there may be an influence of RcsB expression as well (presuming that the site-directed mutants are expressed at higher levels). Furthermore, the K to E mutant in no way mimics lysine acetylation (this is a charge reversal), so the increased motility and flagellar biosynthesis in that site-directed mutant (which is completely the same as seen for the real mimic K to Q) suggests perhaps that lysine SUCCINYLATION (for which K to E is a mimic) is as responsible for the phenotype as acetylation.

10. Authors should use 'sirtuin-like' for describing CobB in the Introduction.

1st Revision - authors' response

07 August 2014

Castaño-Cerezo *et al.* "Protein deacetylation affects acetate metabolism, motility and acid stress response in *Escherichia coli*". Response to reviewers

The authors wish to thank the reviewers for their insightful comments and suggestions, which have greatly contributed to increasing the quality of the manuscript and its overall interest.

Overall, this revised version includes new evidences supporting the roles of protein acetylation in the physiology of *E. coli*. Namely:

- A thoughtful data analysis has demonstrated that protein acetylation of *cobB* and *patZ* mutants in acetate minimal medium is significantly different, and differentially acetylated proteins might explain the different phenotypes exhibited by the mutants.
- A careful inspection of the functions of acetylated proteins and their relative abundances in the global proteome has demonstrated: i) there is a technical bias towards the identification of more abundant proteins and ii) the functions of acetylated proteins found in our assays are a mirror of their abundance in the global proteome.
- New experiments have demonstrated that CobB deacetylates acetyl-CoA synthetase and isocitrate lyase, identifying which lysine residues are specifically targeted by this deacetylase.
- Moreover, the main text, description of methods, the figures, display of datasets and supplementary material of the manuscript have been thoughtfully improved.

Here follows a point by point response the points raised by reviewers.

Reviewer #1:

This an interesting manuscript, that gives new insights into the role of protein acetylation in E. coli. Recently, there has been a great deal of discussion about the enzymatic-vs non-enzymatic lysine-acetylation and the question whether non-enzymatic acetylation still has a function. This manuscript clearly shows cellular functions of lysine-acetylation, validated by site-directed mutagenesis. It goes well beyond previous lysine-acetylome publications in its completeness and follow-up experiments.

The figures supporting the results are unfortunately of a low quality. The fonts are too small everywhere. Combination of red-green should be avoided to help colorblind scientists. There is no mention of which sequences were selected to produce figure 2A.

There is no explanation of colors in figure 2B. Figure 3 has sub-panels A, B, C, E, F (no D) and no caption for F. The metabolic fluxes don't have unit of measurement. How many chemostat cultures were run, is also missing. Figure 5, no explanation of kind of assays is given. Figure 6, no explanation of colors in colorimetric assays, no labels.

Thus my recommendation to majorly revise the figures before a resubmission.

As recommended by reviewer #1, we have increased the quality of the figures and avoided red and green colors. We have also changed figure captions that, in the present version, include more information about the assays (namely: number of replicates, units of data shown and further information about the biochemical basis of the assays).

Regarding former Figure 2 (Figure 3 in the revised version), the following text has been inserted in the caption: "Sequence motif surrounding acetylated lysines. Logo was created using the Icelogo

software package. All acetylated peptides found with an Acetyl (K) Probability higher than 0.9 and a cutoff p-value<0.01 were used.”.

The corresponding section of Materials and Methods has also been updated (Lines 553-555). “Lysine acetylation motif was created using the Icelogo software package. For the generation of the acetylation motif all acetylated peptides with an Acetyl (K) Probability higher than 0.9 and a cutoff p<0.01 were used.”.

Reviewer #2

1. Deletion of patZ caused an increase in global acetylation (Fig. 1D) in E. coli cultured in acetate containing media. This increase is at least as prominent, if not more, as seen in CobB deletion mutants cultured in different conditions (Fig. 1E-H). This suggests that in acetate cultures, patZ suppresses global acetylation to a greater degree than CobB. Strikingly, however, the authors do not observe a strong growth phenotype in PatZ mutants under these conditions, and they clearly acknowledge this as "Despite the high impact of patZ deletion on protein acetylation, no evident phenotypic effects were observed." This indicates that the global increase in acetylation was not sufficient to explain the phenotype of the CobB strain. It raises questions as to whether CobB regulates acetylation of a different set of sites than PatZ to regulate biological functions, and whether all phenotypic effects observed in CobB are related to its deacetylase activity?

Currently, the authors conclude that the phenotypes in CobB are related to its deacetylase function; however, it has not been demonstrated directly. To address point, the authors should generate a mutant strain expressing a deacetylase inactive mutant of CobB, and compare its effects on global acetylation and E. coli phenotype to CobB deletion mutants. These experiments will address whether CobB affects bacterial physiology through its deacetylase activity, or whether CobB has deacetylase independent functions in cells.

Additionally, it would be interesting to know how CobB targets were affected in PatZ deficient cells grown in acetate containing media.

We agree with Reviewer #2. In the previous version of the manuscript, proteomic results were not sufficiently discussed. It is clear from physiological parameters (mainly, growth and metabolite profiles) that the *cobB* mutant is more affected than the *patZ* mutant in acetate cultures. However, protein acetylation is more abundant in the *patZ* mutant than in the wild type strain and the *cobB* mutant.

The biggest phenotypic effects observed in the *cobB* mutant, are presumably due to its inability to consume all acetate in the medium because of the inactivation of acetyl-CoA synthetase. In addition, we have proposed that the more altered phenotype observed in the *cobB* mutant is caused by its specific function as deacetylase, controlling the physiology of *E. coli* through the level of acetylation of relevant proteins. The higher acetylation levels observed in the $\Delta patZ$ mutant are caused by a deregulation of chemical acetylation, which may not alter protein functions i) because of its lower specificity and ii) due to the presence of an active CobB, deacetylating those proteins which are its true substrates. In order to make these observations rational, we have analyzed the proteins that are highly acetylated in each mutant under the acetate culture condition.

The \log_2 of acetylation ratios of each acetylated peptide in $\Delta patZ$ mutant were plotted versus the \log_2 acetylation ratio of the same peptide in $\Delta cobB$ (Figure 2A). This plot evidences that a big portion of the peptides acetylated in acetate cultures have high acetylation ratios in both mutants. This suggests that both proteins are responsible for the control of protein acetylation in this condition. However, a group of peptides are significantly more acetylated in the *patZ* or the *cobB* mutant (two sample t-test, adjusted for multiple testing using permutation based FDR<0.05) (Fig. 2B). Thus, the set of proteins, which have a high acetylation ratio in the *cobB* mutant but lower in the *patZ* mutant, is likely enriched in true substrates of CobB. On the other hand, the increase in protein acetylation in the *patZ* mutant is due to the overall increase of chemical acetylation, being non-substrate specific, and quite likely, true substrates of CobB are under represented in this dataset. Moreover, proteins which are more acetylated in the *cobB* mutant are likely exerting the phenotypic effects observed. To support this hypothesis, we have selected a few examples of proteins whose lysine residues targeted by acetylation are likely involved in an alteration of protein functionality and, therefore, could contribute to the altered phenotype of the mutant (Fig. 2C). The peptide acetylation ratios of the following three proteins were significantly different in the *cobB* mutant.

- Lysine 269 of fructose-1,6-bisphosphatase class 1 (Fbp). This is the clearest example. This lysine is highly conserved and in previous studies it was demonstrated that the modification of this residue at active site to an alanine reduces the affinity of this enzyme for its

substrate 20-fold, and decreases affinity for the competitive inhibitor fructose 2,6-bisphosphate 500-fold (el-Maghrabi *et al*, 1992). We postulate that the higher acetylation of this residue in the $\Delta cobB$ mutant might alter Fbp activity and, therefore, gluconeogenesis.

- Lysine 557 of the RNA polymerase sigma subunit, RpoD is more acetylated in the *cobB* mutant. This lysine belongs to a HTH domain responsible for the interaction with the -35 box of gene promoters. It is not clear if this residue interacts directly with the DNA backbone, but the abundance of positively charged amino acids in this domain, suggests that it may play a role in the fine tuning of transcription initiation in *E. coli* (Paget & Helmann, 2003).
- The ribosome-associated inhibitor A protein (Rai) regulates the translation in stationary phase in *E. coli*. It can block the dimer formation of 70S, decreasing the translation rate. This protein has two different regions that block the A-site (aminoacyl-tRNA site) and P-site (peptidyl-tRNA site) of the ribosome blocking translation. Both regions are particularly enriched with positively charged amino acids. The acetylation of lysine 87, which belongs to the second region, suggests that Rai function could be compromised in the *cobB* mutant (Agafonov *et al*, 1999; Maki *et al*, 2000).

Apart from these proteins whose acetylation ratios were significantly different in the *cobB* mutant, acetylation of other proteins was also detected in some of the biological replicates. They are described below.

- The long chain fatty acid CoA ligase (FadD) is acetylated in its active site, as occurs on Acs, the acetylation ratio was 35-fold higher in the *cobB* mutant than in the *patZ* mutant. Regulation of FadD by lysine acetylation has also been described in other organisms like *Rhodospseudomonas palustris* (Crosby *et al*, 2010).
- The glyceraldehyde-3-phosphate dehydrogenase (GapA) was found to be more acetylated only in the *cobB* mutant in lysine 184 (Kuhn *et al*, 2014). The acetylation of this residue alters NAD⁺ binding and, therefore, its activity. It has been demonstrated that the activity of GapA is essential under glycolytic and gluconeogenic conditions (Seta *et al*, 1997) and, therefore, its inactivation could alter $\Delta cobB$ physiology.

The following paragraph was inserted in the Main Manuscript:

Revised Text: (Lines 158-176): “The intriguing accumulation of acetylated proteins in acetate cultures was further analyzed. About 15% of peptide acetylation ratios were significantly different in the two mutants (Figure 2A). Statistical significance levels were determined by two sample t-test, followed by multiple testing correction using permutation-based FDR<0.05 (Tusher *et al*, 2001). These differences in the protein acetylation profiles of the *patZ* and *cobB* mutants in acetate mirror the different degree of phenotype alteration observed. To get an insight on which of the acetylated proteins might be responsible for the different phenotypes we focused on those with high acetylation ratios only in the *cobB* mutant. All acetylated peptides with a ratio 2-fold higher than the wild type were analyzed; of these, 25 peptides were identified as highly acetylated only in the *cobB* mutant but not in the *patZ* mutant (two sample t-test, adjusted for multiple testing using permutation based FDR<0.05)(Figure 2B, Supplementary dataset 2). By manual curation, proteins which are acetylated at lysine residues whose have been previously identified as relevant for their function were identified in this group (Figure 2C; Supplementary table 2). These proteins could contribute to growth impairment in the *cobB* mutant. On the other hand, the higher acetylation levels observed in the $\Delta patZ$ mutant were probably caused by a deregulation of chemical acetylation. This may not alter cell growth significantly due to i) its lower specificity (probably affecting only a fraction of the total cellular protein) and ii) due to the presence of an active CobB, which would deacetylate those proteins which are its true substrates, and which acetylation state is really crucial for cellular functions.”

As proposed by reviewer #2, an inactive CobB protein was constructed by mutating histidine 110 in the active site of the protein for a tyrosine, as previously described by other authors (Borra *et al*, 2002; Finnin *et al*, 2001). We performed complementation experiments using two different constructions (pBAD24-*cobB* and pBAD24-*cobBH110Y*) transformed into *E. coli* $\Delta cobB$ mutant. In these experiments we have observed that the overexpression of the native CobB is lethal in minimal media while the growth of cells carrying the H110Y mutant construction is similar to the $\Delta cobB$

mutant (Suppl. Fig. 4). Therefore, this experiment further supports that the phenotypic effects observed in response to CobB deletion are due to its deacetylase activity and rules out that these effects could be due to a deacetylase independent activity in the cell.

Revised text (Lines: 146-157): “Deletion of *cobB* caused substantial phenotypic changes. The growth rate of this mutant was reduced in all conditions assayed (Table 1). Deletion of the only deacetylase known in *E. coli* should increase the degree of acetylation of proteins. Our results confirmed that CobB has a major role as deacetylase in *E. coli*. Over 17% of the acetylated peptides quantified showed increased acetylation under each of the chosen growth conditions (at least 2-fold in the $\Delta cobB$ mutant compared to the wild type) (Figure 1E-H). The number of peptides with increased acetylation was higher in the conditions where the change in phenotype was more profound, *i.e.* acetate and chemostat cultures (30% and 21% respectively). Since the expression of an inactivated CobB protein, with a mutation in its catalytic H110 residue did not rescue the phenotype of the *cobB* knockout mutant, we concluded that the phenotypic and proteomic effect observed in this mutant is caused by the absence of the deacetylase activity (Supplementary Figure 4).

2. The authors perform functional analysis of acetylated proteins (Page 6, Figure 2B) and conclude that acetylation occurs more frequently on metabolic proteins, and that metabolic proteins are often multiply acetylated. However, proteome-wide studies of acetylation are biased towards abundant proteins and the frequency of observed acetylation is correlated to protein abundance. Many proteomic studies of acetylation have made the mistake of misinterpreting the technical bias to detect acetylation on highly abundant metabolic proteins to indicate that metabolic proteins are more highly acetylated than other proteins, the authors should avoid making the same mistake. The results shown in Figure 2B may simply reflect the abundance of these proteins. It would be more appropriate to correct acetylation abundance bias to protein abundance, and repeat these analyses. At a minimum the authors should indicate that these results could be attributed to the abundance of metabolic proteins or the analysis should be removed altogether.

We agree with reviewer #2 that protein abundance (*i.e.* its actual concentration in the cell) affects the probability of detecting their acetylated peptides and, therefore, our acetylated peptides dataset might be biased towards abundant proteins. In order to test this, for each condition and biological replicate all proteins detected in the proteomic studies were grouped into five groups or quantiles depending on their intensities (where quantile 1 groups the less abundant proteins and quantile 5 the most abundant ones). Protein relative quantification was based on LC-MS runs with tryptic digests of whole protein extracts (*i.e.*, prior to immunoprecipitation of acetylated peptides). Subsequently, acetylated proteins were classified into the different proteome quantiles. The percentage of acetylated proteins in each quantile is represented in Fig. 3B. It is evident that, as suggested by Reviewer #2, most of the acetylated proteins that have been identified in our study belong to proteins that are highly abundant in the proteome, *i.e.* quantiles 4 and 5 (Fig 3B). Therefore, the detection of proteins acetylated is evidently biased by the abundance of the proteins in the whole proteome.

In order to test if our previous Gene Ontology analysis was biased by protein abundance, we analyzed the different biological functions represented by proteins belonging to all quantiles. The abundance of the different GO terms was similar in each quantile, meaning that functions defined by GO terms are evenly distributed across the whole dynamic range of protein abundances. Moreover, these frequencies of GO terms were very similar to the frequencies at which they are represented in the whole proteome (Supp. Fig. 5).

Therefore, we can conclude that i) detection of protein acetylation is a function of protein abundance in the cell and ii) the frequency of GO terms in the set of acetylated proteins reflects their own frequency in the whole genome, and no specific function is over-represented (Supp. Fig 5).

The text of the Main Manuscript has been modified as follows.

Previous text:” The analysis of the functions of acetylated proteins sheds light on the major biological processes affected. In our study, 64% of the modified proteins detected have a metabolic function, and almost 80% of these are involved in primary metabolism, for example nucleotide and amino acid biosynthesis and carbohydrate metabolism (Fig. 2B). Other overrepresented functions in our data set relate to sensing and stimulus responses, mostly proteins belonging to two component systems, such as ArcA, RcsB, CpxR and EvgA. Additionally, almost 7% of the acetylated proteins have a role in transcription. Altogether, these results show that lysine acetylation, although very

prominent in metabolism, has likely multiple biological functions that include cell communication, stress survival and transcriptional regulation (Suppl. Table 5)."

Revised text (Lines: 186-197): "Importantly, detection of acetylated peptides is affected by protein abundance in the cell since minority proteins are likely out of the reach of current techniques. Therefore, the subset of acetylated proteins detected is biased towards those proteins which concentration in the cell is high (Figure 3B, Supplementary material). The analysis of the functions of acetylated proteins (Gene Ontology terms) sheds light on the major biological processes affected. In our study, 64% of the modified proteins detected have a metabolic function, and almost 80% of these are involved in primary metabolism, for example nucleotide and amino acid biosynthesis and carbohydrate metabolism (Supplementary Figure 5). Other represented functions in our dataset relate to sensing and stimulus responses, mostly proteins belonging to two component systems, such as ArcA, RcsB, CpxR and EvgA. Additionally, almost 7% of the acetylated proteins have a role in transcription (Supplementary Table 3). However, the frequency of these GO terms in the set of acetylated proteins reflects their own frequency in the whole genome, which means that no specific function is over-represented in the group of acetylated proteins and reveals that protein acetylation occurs on every type of protein independently of its function (Supplementary Material)." An extended explanation of this study is included in the Supplementary Materials section.

3. Their more novel finding, that CobB regulates isocitrate lyase through deacetylation, is not supported well-enough. The authors find that isocitrate lyase (Icd) protein abundance was slightly reduced (~50%) in CobB-deficient bacteria, however, Icd activity was ~20-fold reduced in cell free extracts from CobB-deficient cells grown in chemostat culture. The authors conclude that acetylation fine-tunes Icd activity. However, the authors were unable to detect altered Icd activity after treating purified Icd with CobB, which they explain may be due to the purified enzyme being present in an active, non-acetylated form. In order to provide better experimental evidence that CobB regulates Icd through direct deacetylation, rather than by affecting its protein levels or through an unknown mechanism, the authors should purify Icd under conditions where its activity is affected, presumably, by acetylation and then treat the purified protein with CobB to test whether this treatment rescues Icd activity. For example, using Icd purified from CobB-deficient cells grown in chemostat culture.

The panels in Figure 3 are not labeled correctly (E and F, versus D and E in the text and figure legend). In addition the text states that "isocitrate lyase and isocitrate dehydrogenase levels were measured (Fig. 3D and 3E)" while the figure legend indicates "Enzyme activities of (B)Acs, (D)AceA, and (E) Icd..." the gene symbols should be defined in either the text or the figure legend, or both. This could be made even clearer by indicating the enzyme activity being measured on the axis of the column chart shown in the figure itself.

We agree with reviewer #2 that in the previous version of the manuscript we lacked sufficient experimental data supporting the effect of acetylation on isocitrate lyase activity. As proposed by reviewer #2, we have purified AceA protein in those conditions where its enzyme activity and the corresponding flux through the glyoxylate shunt were decreased. For that aim, we transformed the *E. coli* K12 $\Delta cobB$ mutant with ASKA AceA plasmid (pCA24N-*aceA*) and we grew it in minimal medium with acetate as sole carbon source. Cells were harvested when cultures reached OD=0.3 and AceA was purified using IMAC chromatography (further described in the Material and Methods section). The *in vitro* deacetylation assays were carried out on purified AceA. Around 40% increase in AceA activity was observed upon treatment with purified CobB protein, compared to a control without CobB. This effect was partially inhibited by nicotinamide. Partial deacetylation of AceA was confirmed by Western Blotting and LC-MS/MS. We have found out that CobB deacetylates lysines 13 and 308 extensively, a reaction which is inhibited by nicotinamide. This result is in agreement with previous finding in *S. enterica*. It was demonstrated that the acetylation of lysine 308 decreases AceA activity (Wang *et al.*, 2010).

After these results we have included several changes in Fig. 3. Enzyme activities of isocitrate dehydrogenase (Icdh), isocitrate lyase (AceA), acetyl-CoA synthetase (Acs), phosphotrasacetylase (Pta) and acetate kinase (AckA) in crude cell extracts, have been transferred to Suppl. Material (Suppl. Fig 5 in current version). Novel data of AceA activity after *in vitro* deacetylation and its verification through Western Blotting and the relative intensity of the acetylated peptides of control and CobB treated AceA have now been included in this figure. Additional data on the deacetylation of acetyl-CoA synthetase by CobB have also been included.

Previous text: “*In vitro* isocitrate lyase deacetylation assays were carried out but the enzyme activity was not altered after incubation with CobB, potentially because the enzyme was purified activated (*i.e.*, in the non-acetylated form).”

Revised text (Lines 263-268): “*In vitro* isocitrate lyase deacetylation assays showed that after the incubation with the sirtuin-like deacetylase CobB, the activity of AceA increased around 40% (Fig. 4D). We have found out that CobB deacetylates lysines 13 and 308 extensively, a reaction which is inhibited by nicotinamide (Fig. 4E). This result is in agreement with previous finding that the acetylation of lysine 308 decreases AceA activity in *S. enterica* (Wang et al, 2010).”

4. The authors make the interesting finding that acetylation levels are elevated in PatZ-deficient bacteria grown in acetate media. In the discussion they present a reasonable explanation that this may be due to elevated activity of Acs in PatZ-deficient cells, resulting in elevated acetyl-CoA levels and therefore globally elevated acetylation. The manuscript would be strengthened by performing some fairly straightforward experiments to test this theory. The authors could assay Acs activity (which they have done already in this study) in PatZ-deficient cells grown in acetate media, compared with other growth conditions and wild-type cells. They could further compare acetyl-CoA levels between these strains and growth conditions. These experiments would strengthen the manuscript by providing further insight into the only remarkable finding attributed to the loss of PatZ.

We agree with Reviewer #2 that one of the most relevant contributions of our work is the establishment of a link between PatZ and the control of chemical acetylation. The higher protein acetylation observed in the *patZ* mutant could be caused by the increased production of acetyl-CoA or acetyl-P. Some authors, such as Weinert and collaborators and others, have already demonstrated that increased production of acetyl-P or acetyl-CoA leads to higher protein acetylation in *E. coli* and yeast.

In *E. coli*, during growth on acetate, the enzymes responsible for the metabolism of acetylating agents are: acetyl-Coenzyme A synthetase (Acs), phosphotransacetylase (Pta) and acetate kinase (AckA). We have measured the activities of the three enzymes in cell crude extracts (Suppl. Fig. 5) in the different strains and growth conditions. We observed that two of these activities, Acs and Pta, were higher in acetate cultures in $\Delta patZ$ compared with the wild type strain (two-way Anova, $p < 0.0001$). In this work we have demonstrated that acetylation of Acs inhibits the activity of this enzyme (Figure 4). Moreover, we have also demonstrated the acetylation of Pta in our experimental dataset, more precisely in lysine 317. Although we have not analyzed the effect that acetylation of this residue has on the activity of Pta, our data indicate a correlation between the acetylation ratio of this site and the activity of this enzyme in the mutants ($\Delta cobB$, $\Delta patZ$) and the parent strain. In all conditions, except for one, an increase on acetylation led an increase in the enzyme activity. Moreover, other authors have previously described that both Acs and Pta are highly acetylated in the proteome of *E. coli*. These authors investigated the stoichiometry of lysine acetylation in *E. coli* and they observed that most of the proteins that use acetyl-CoA as substrate or product (as in the case of Acs and Pta) are more abundantly acetylated than the average (Baeza *et al*, 2014). One of these lysine residues which are “abundantly acetylated” is lysine 317 of Pta. Altogether, increased activities of Acs and Pta in $\Delta patZ$ suggest a higher capability for the synthesis of the acetylating agents acetyl-CoA and acetyl-P.

The steady state levels of coenzyme A thioesters could also shed light on this. Weinert *et al.* (2014) observed changes in the acetyl-CoA pool in yeast after deletion of the major mitochondrial pathways pyruvate dehydrogenase and citrate synthase, which correlated with the accumulation of acetylated proteins. To further show if acetyl-CoA levels could be responsible for the increase in protein acetylation in *E. coli* $\Delta patZ$, the intracellular concentration of coenzyme A, acetyl-CoA and succinyl-CoA was measured in the $\Delta cobB$ and $\Delta patZ$ mutants and the parental strain in acetate cultures (Suppl. Fig 10). The biggest differences were observed in the $\Delta cobB$ mutant, where the levels of acetyl-CoA and CoA were half of those of the wild type. Nevertheless, the acetyl-CoA levels in the $\Delta patZ$ mutant were not significantly different to those of the control. However, changes in the acetyl-CoA production rates in the cell could be “buffered” due to i) the big number of enzymes that use this metabolite as substrate, ii) because of its use as protein acetylating agent, and iii) because increased production of acetyl-CoA might lead to the accumulation of acetyl-P. Therefore, acetyl-CoA levels in the steady state could remain close to those of the control even if the net rate of acetyl-CoA synthesis was higher. Thus, in our view, this experimental observation does not rule out a connection between altered acetyl-CoA production and protein acetylation in *E. coli*.

Altogether, increased activities of Acs and Pta in the *patZ* mutant suggest a higher capability for the synthesis of the acetylating agents acetyl-CoA and acetyl-P, which could lead to increased acetylation of proteins.

Inserted text (Lines 384-385): “However, deletion of *patZ* did not alter the pool of acetyl-CoA, succinyl-CoA and free CoA (Suppl. Fig. 10).”.

Minor concerns

1. *The reviewing a manuscript could be made much easier by providing both page numbers, line numbers, and labeling figures with the figure number.*

Page and line numbers have been inserted. Figures are inserted in the final PDF by the online submission system of MSB.

2. *Acetylation has been known to occur since the 1960s (the authors indicate 1970s)*
Corrected.

3. *The iceLogo comparison is not sufficiently explained in the methods section. The authors should indicate what the acetylated sequences were compared to?*

Lysine acetylation motif was created using the Icelogo software package. For the generation of the acetylation motif all acetylated peptides with an Acetyl (K) Probability higher than 0.9 and a cut-off $p < 0.01$ were used.

4. *All MS results should be provided in Excel spreadsheets for easier navigation of the results.*
Done.

5. *Fig. S3B, the authors note a decreased acetylation of K609 on Acs in PatZ cells. However, acetylation also appears somewhat reduced in CobB, which appear inconsistent with this site being deacetylated by CobB.*

We also noticed that the intensity of this peptide in the *cobB* mutant and *wt* strain is similar. But after normalization of peak intensity with the protein level a higher acetylation ratio is observed in the *cobB* mutant (Acs protein ratio *cobB/wt* = 0.43).

6. *The authors indicate "Flux through the glyoxylate shunt decreased by 50% in the....." However, the Fig. 3C, seems to indicate that the flux decrease was ~33%?*
Corrected.

Previous text: To further explore the functional consequences of the acetylation of metabolic enzymes, intracellular fluxes were determined by ^{13}C -experiments (Suppl. Table 1). Flux through the glyoxylate shunt decreased by 50% in the ΔcobB mutant in glucose chemostat cultures (Fig 3C).
Revised Text (Lines 245-248): “To further explore the functional consequences of the acetylation of metabolic enzymes, intracellular fluxes were determined by ^{13}C -experiments (Supplementary Table 1). Flux through the glyoxylate shunt decreased by 34% in the ΔcobB mutant in glucose chemostat cultures (Fig 3C).”

7. *Fig. S5 appears redundant with Fig. 4A, as the Fig. S5 does not provide information about the clustered terms.*
Former Figure S5 has been deleted.

8. *Fig. 6, the authors should clearly explain how to interpret the provided images. The readership of MSB is not necessarily familiar with these types of analyses and some explanation would help them better understand the provided data.*

In the current version, we have improved Caption to Figure 6 (Figure 7 in the revised version), including further data on the experimental methods used to allow the readers to understand the basis of physiological experiments: “Fig.6. Effect of RcsB inactivation on acid stress response. Glutamate decarboxylase (Gad) enzyme activity. Colorimetric enzyme activity assay was followed by the consumption of H^+ increasing the pH of the reaction *in vitro*. The increase of pH was detected by the color turn from yellow to blue of the pH indicator bromocresol green (A) Acid stress survival of the different *E. coli* mutants. The *rcsB* knockout mutant was complemented with the *rcsB* wild type gene (K154) and its different mutants mimicking different acylation states of lysine 154 (K154R, K154Q and K154E). Bacteria were grown overnight in minimal medium with pH 5.5 to pre-adapt them to growth at low pH. These bacteria were subjected to acid stress for 2 hours (pH 2.5) and cell survival was measured afterwards (B).”.

Reviewer #3:

1. For the mapping experiments of lysine acetylated peptides, the authors state that 4 biological replicates were used, but do not state the basis for how the data from each replicate were included. Was a peptide included if it was identified once in one biological replicate? What confidence thresholds were applied (scoring parameters etc.)? How were site allocations defined? In the quantification studies, were peptides only confidently quantified if they were identified in 1,2,3 or 4 experiments, and what statistical tests were performed? I think in summary, substantially more information may be required here.

We have completed the information given in the previous version of the manuscript. Data analysis was carried out using MaxQuant (version 1.3.0.5). For statistical analysis we used Perseus (1.3.0.4). Stringent confidence parameters were used for the identification of acetylation sites (see below and information on experimental procedures, data analysis). Answering to the questions of the reviewer:

a) Occurrence of peptides in biological replicates and statistical analysis of data. For quantification of peptide acetylation ratios, all peptides were used, without considering whether they were only present in one or several replicates. The median of those values was used for the construction of the frequency histograms (e.g. Figure 1 and Suppl. Figure 3). as it is reported in other proteomic studies (Weinert *et al*, 2013) For in depth analysis of significantly affected peptide acetylation ratios (e.g. Fig. 2 and Suppl. Data 2) only those peptides found in two or more replicates were used. Significance of differences found in the median of peptide acetylation ratios was assessed statistically, and only those values with an FDR<0.05 (two sample t-test, adjusted for multiple testing using permutation based FDR<0.05) were considered.

b) Peptide identification and definition of site allocations. Raw data was analysed by MaxQuant (version 1.3.0.5) (Cox & Mann, 2008). Andromeda (Cox *et al*, 2011) was used to search the MS/MS data against the Uniprot *E.coli* MG1655 database (version v2012-09, 4431 sequences), including a list of common contaminants and concatenated with the reversed version of all sequences. Trypsin/P was chosen as cleavage specificity allowing three missed cleavages. Carbamidomethylation (C) was set as a fixed modification, while Oxidation (M), Acetyl (Protein N-term) and Acetyl (K) were used as variable modifications. For dimethyl labelling DimethylLys0 and DimethylNter0 were set as light labels, DimethylLys4 and DimethylNter4 were set as medium labels, and DimethylLys8 and DimethylNter8 were set as heavy labels. The database searches were performed using a peptide tolerance of 20 ppm for the first search and 6 ppm for the main search. HCD fragment ion tolerance was set to 20 ppm. Data filtering was carried out using the following parameters: Peptide False Discovery Rate (FDR) was set to 1%; Andromeda score was set to 30; max peptide PEP was set to 1; minimum peptide length was set to 5; minimum razor peptides were set to 1; peptides used for protein quantification was set to razor and unique peptides; Protein quantification was performed by using only unmodified peptides and Oxidation (M) and Acetyl (Protein N-term); the re-quantify option was enabled. Acetylated lysines were localized as previously in Olsen J.V. *et al*, 2010(Olsen *et al*, 2010). Only peptides, with a localization probability higher than 0.5, were reported.

Revised text (Lines: 490-509): “Raw data was analysed by MaxQuant (version 1.3.0.5) (Cox & Mann, 2008). Andromeda (Cox *et al*, 2011) was used to search the MS/MS data against the Uniprot *E.coli* MG1655 database (version v2012-09, 4431 sequences), including a list of common contaminants and concatenated with the reversed version of all sequences. Trypsin/P was chosen as cleavage specificity allowing three missed cleavages. Carbamidomethylation (C) was set as a fixed modification, while Oxidation (M), Acetyl (Protein N-term) and Acetyl (K) were used as variable modifications. For dimethyl labelling DimethylLys0 and DimethylNter0 were set as light labels, DimethylLys4 and DimethylNter4 were set as medium labels, and DimethylLys8 and DimethylNter8 were set as heavy labels. All peptides were used for quantification studies, but to find significantly acetylated peptides (Figure 2), only those found in two or more replicates and with a FDR<0.05 were used (t-test adjusted for multiple testing using permutation based FDR). The database searches were performed using a peptide tolerance of 20 ppm for the first search and 6 ppm for the main search. HCD fragment ion tolerance was set to 20 ppm. Data filtering was carried out using the following parameters: Peptide False Discovery Rate (FDR) was set to 1%; Andromeda score was set to 30; max peptide PEP was set to 1; minimum peptide length was set to 5; minimum

razor peptides were set to 1; peptides used for protein quantification was set to razor and unique peptides; Protein quantification was performed by using only unmodified peptides and Oxidation (M) and Acetyl (Protein N-term); the re-quantify option was enabled. Further data processing was performed using the Perseus tool (version 1.3.0.4) available in the MaxQuant environment.

2. Results, p.2, 'Deletion of *cobB* caused substantial phenotypic changes' and then all following experiments for example the flux experiments etc. - these studies would all be strengthened by the inclusion of a *cobB* complement (as above).

As proposed by reviewer #3, we have tried to complement the deletion of *cobB* by expressing this gene *in trans* from a expression vector (pBAD24). However, although *E. coli* BW25113 $\Delta cobB$ cells were successfully transformed with the plasmid pBAD*cobB* and were capable of growing with this plasmid in complex (Lysogeny Broth) medium, we were unable to grow it in minimal acetate medium.

Moreover, this deleterious effect was due to the deacetylase activity of *cobB*, since a H110Y mutant (which, according to literature, would lack deacetylase activity) was able to grow under the same environmental conditions.

We have included this additional information in the Supplementary Material: Supplementary Information and Supplementary Figure 4.

3. When using the anti-lysine acetylation antibody and performing quantitation, were the unbound proteins / peptides also analysed to control for non-specific binding and loss of acetylated proteins? It is generally useful to determine whether whole protein abundances change under chosen conditions. The authors use transcript arrays to examine the genomic response to *cobB* deletion but they most likely also have access to the proteome-level data. These should be included.

In this study we have checked acetylation enrichment after immunoprecipitation (IP) but we did not check the loss of acetylated peptides after the immunoprecipitation. In the current version of the manuscript we have included, as proposed by Reviewer #3, another file (Suppl. data 3) with the relative acetylation of each of the sites (AcK ratio/ Protein ratio). In the previous version of the manuscript the ratios supplied in the text were already corrected by protein abundance.

We also have compared the proteomic and transcriptomic data in chemostat cultures in both mutants. We observed that the correlation between them is nice, especially for the *cobB* mutant where the changes were more severe (Pearson correlation coefficient=0.72). These data were not included in the manuscript, since no further useful information can be obtained from them.

4. Results, p.1 '...in exponential phase the *patZ* gene expression is low...' Did the authors confirm this or is this previously published? Also '...might be explained by the presence of at least 25 putative acetyltransferases...' - another reason to examine unmodified protein abundances to determine whether the abundance of any of these is altered by *patZ* deletion.

Across the manuscript we state that *patZ* gene expression is high at stationary phase and acetate cultures because its gene expression was extensively studied in a previous paper (Castaño-Cerezo et al, 2011 Mol Microbiol 82(5): 1110-1128). In that paper it was demonstrated that the transcription of *patZ* was regulated by catabolite repression, its expression being high at the stationary phase and acetate cultures but not in glucose exponential phase.

We agree with reviewer #3 that another of the 25 putative acetyltransferases annotated in *E. coli* might contribute to the changes observed in acetylation, however, we have not found data, which might clearly suggest this point. We have also checked, using microarray data, the relative expression of 25 N-acetyltransferases (with known or unknown function). We observed that some of these N-acetyltransferases have their expression altered in the *patZ* mutant, but also in the *cobB* mutant in chemostat cultures, while the differences of expression of these genes in glucose exponential phase were almost negligible. Moreover, some of these N-acetyltransferases, such as *speG* and *argA* are acetylated, but their change in gene expression and protein acetylation cannot be described as altered due to the absence of PatZ. Undoubtedly, the roles of these acetyltransferases in the regulation of acetylation of proteins are worth of further study.

These data were not included in the manuscript, since no further useful information can be obtained from them.

5. Results, p.2, 2nd para - if CobB is the major deacetylase, what is the explanation for the (relatively) low proportion of acetylated peptides influenced by *cobB* deletion (17%-up to 30% in acetate)? Furthermore, in the Results on p.3, para 3, the authors state that 8 acetylation sites on Acs were unaffected by *cobB* deletion. So in some respects the results indicate that CobB is not the major deacetylase, but perhaps one of two or more deacetylases?

We agree with Reviewer #3. We assert along the manuscript that CobB is the main/only deacetylase known in *E. coli*, but the existence of other non-identified deacetylases cannot be discarded. There are many acetylation sites that do not seem to be regulated by CobB. This could be due to the presence of other deacetylases in *E. coli* not found yet. This might mean that the rest of acetylation sites might be irreversibly acetylated. Moreover, the functional relevance of the modified site and the abundance of each acetylation in the proteome (*i.e.* which fraction of the total protein population in the cell is actually acetylated) must be taken into consideration. In our data set we have 9 acetylation sites for acetyl CoA synthetase. Most of them have no alteration in their acetylation ratio but, as reported by Baeza et al, only one of these acetylation events is abundant in the proteome. Probably those 7 sites (excluding 609 and 307) are low abundant and probably irreversibly acetylated.

6. Results, 3rd para - a motif analysis of CobB substrates is presented in Weinert et al.

Reviewer #3 is right. The acetylation motif reported by Weinert et al. (2013) is similar to ours. We have added a citation to this work in the text.

7. How specific in *E. coli* is nicotinamide to CobB? Nicotinamide in eukaryotes has substantial off-target effects other than inhibition of Sirts, for example, on NAD-dependent enzymes. This should be addressed.

We agree with Reviewer #3 that nicotinamide might inhibit many enzymes other than sirtuins, especially NAD⁺-binding enzymes. Several previous lysine acetylation studies in higher and prokaryotic organisms have used nicotinamide as CobB inhibitor (Starai et al, 2004; Gardner & Escalante-Semerena, 2009; Wang et al, 2010; Zhilhoung et al, 2011; Nambi et al, 2013). However, it should be kept in mind that we have only used nicotinamide to inhibit CobB in *in vitro* assays with purified proteins. Moreover, none of the proteins used for these experiments (acetyl-CoA synthetase and isocitrate lyase) are NAD⁺-binding proteins. Moreover, the effect of nicotinamide on the acetylation of these proteins has been demonstrated by western blotting in which we specifically target the deacetylase activity.

The use of this inhibitor as a control for *in vivo* experiments would be much more problematic, given the presumed lack of specificity. In our work, we have not performed any *in vivo* inhibition of CobB using nicotinamide in the culture. Therefore, we do not think that this might be an issue for our manuscript.

8. As an additional control, does nicotinamide (or other sirt/CobB inhibitor) influence motility and / or the expression of flagellar genes?

See previous answer.

9. The studies specifically examining the lysine 154 residue in RcsB are a highlight of the study, but something doesn't add up. While clearly loss of CobB increases flagellar biosynthesis and motility (A/B1-2), mimicking the non-acetylated lysine appears to alter the cell morphology, and the motility is increased relative to wild-type. This suggests there may be an influence of RcsB expression as well (presuming that the site-directed mutants are expressed at higher levels).

Furthermore, the K to E mutant in no way mimics lysine acetylation (this is a charge reversal), so the increased motility and flagellar biosynthesis in that site-directed mutant (which is completely the same as seen for the real mimic K to Q) suggests perhaps that lysine SUCCINYLYATION (for which K to E is a mimic) is as responsible for the phenotype as acetylation.

We agree with reviewer #3: the K to E mutation does not mimic the acetylation of residue 154 but, rather, its succinylation. In the current version of the manuscript we state this fact.

Revised text (Lines: 336-337): "Shifting this residue to a negatively charged glutamate (K154E) would mimic a succinylated lysine."

The reason behind this mutation was to determine the effect of switching the electrostatic charge at position 154 of transcription factor RcsB. In fact, altogether, K to Q, K to R and K to E mutants demonstrate that positively charged side chain of K154 is essential for DNA binding, indicating that any post-translational modification affecting local charge at this residue would yield a similar effect. While in this work we have detected the acetylation of K154, this residue could also be subjected to other posttranslational modifications affecting local electrostatic charge, including succinylation. However, that is clearly beyond the scope of this work.

10. Authors should use 'sirtuin-like' for describing CobB in the Introduction.

Corrected.

2nd Editorial Decision

26 September 2014

Thank you again for submitting your work to Molecular Systems Biology. We have now heard back from the two referees who were asked to evaluate your manuscript. As you will see below, the referees think that their main concerns have been satisfactorily addressed. However, reviewer #3 lists some minor comments, which we would ask you to address in a revision of the manuscript. In particular, these comments refer to the need to incorporate the comments on the K154E RcsB mutant in the Discussion section of the manuscript and to include a citation to the study of Colak et al., 2013 (PMID: 24176774).

On a more editorial level, we would like to draw your attention to the following:

- We would like to ask you to deposit the proteomics and microarray data in one of the major public databases and to provide the dataset identifier in the "Data Availability" section of your manuscript.

Reviewer #2:

The authors have adequately addressed my concerns, and the revised manuscript can be recommended for publication.

Reviewer #3:

The authors have provided an updated version of their manuscript, addressing several of the concerns that were raised at original review. I accept that they have attempted to complement cobB and the data presented in the supplement are appropriate. Better discussion of the quantitative aspects of the experiments has also been provided (original reviewer 3, point 1). With respect to the K to E mutant and the mimic of succinylation, it would be appropriate to include the discussion raised by the authors in their rebuttal letter in the manuscript itself. Other points have been dealt

with. The manuscript thus represents a solid body of work, and the presentation is much improved.

I remain unconvinced however, regarding the overall novelty of the work with respect to the published literature and the authors do not deal with some of the major criticisms raised in the original review (e.g functional analysis of isocitrate lyase linked to a specific acetylated lysine). Additionally, why do the authors still refuse to cite Colak et al (MCP 2013)? This work shows the multi-functional nature of CobB for lysine deacetylation and desuccinylation, and thus the role of acetylation / deacetylation in CobB and metabolism remains somewhat blurred.

2nd Revision - authors' response

14 October 2014

The authors wish to thank again all three Reviewers for their constructive criticisms on the manuscript and their suggestions, which we truly believe have contributed to increasing its scientific value.

Here follows a point by point response to the points raised by the Editor and the Referees in this second revision.

#Editor.

1. We would like to ask you to deposit the proteomics and microarray data in one of the major public databases and to provide the dataset identifier in the "Data Availability" section of your manuscript. For more information you can refer you our journal policies on data deposition (<http://msb.embopress.org/authorguide>).

Both proteomics and microarray data have been deposited in public databases. Details and accession numbers have been added to the section Data Availability in the manuscript.

Reviewer #2

The authors have adequately addressed my concerns, and the revised manuscript can be recommended for publication.

Reviewer #3:

1. The authors have provided an updated version of their manuscript, addressing several of the concerns that were raised at original review. I accept that they have attempted to complement cobB and the data presented in the supplement are appropriate. Better discussion of the quantitative aspects of the experiments has also been provided (original reviewer 3, point 1).

With respect to the K to E mutant and the mimic of succinylation, it would be appropriate to include the discussion raised by the authors in their rebuttal letter in the manuscript itself.

Other points have been dealt with. The manuscript thus represents a solid body of work, and the presentation is much improved.

Authors. The authors thank the reviewer for his comments on the review. The discussion on the K to E mutant of RcsB and the possible roles of succinylation has now been included in the discussion section of the manuscript.

2. I remain unconvinced however, regarding the overall novelty of the work with respect to the published literature and the authors do not deal with some of the major criticisms raised in the original review (e.g functional analysis of isocitrate lyase linked to a specific acetylated lysine). Additionally, why do the authors still refuse to cite Colak et al (MCP 2013)? This work shows the multi-functional nature of CobB for lysine deacetylation and desuccinylation, and thus the role of acetylation / deacetylation in CobB and metabolism remains somewhat blurred.

Authors. We do believe that our work contributes to the State of the Art of physiological roles of protein acetylation in *Escherichia coli*. This is a novel field and there is still a high controversy with actual roles and effects of these protein modification mechanisms. Although several authors have aimed at identifying acetylation targets, few physiological studies have been conducted. Here we integrated proteomics with transcriptomics, fluxomics and metabolic data, which allowed us to couple our proteomic observations with physiological effects exerted. The citation to Colak et al (2013) was not added to the previous version by mistake. We apologize for that. It has been now included in the main manuscript.